# Sequence features of retrotransposons allow for epigenetic variability

**Kevin R Costello[1,2], Amy Leung[1], Candi Trac[1], Michael Lee[1,2], Mudaser Basam[1], J Andrew Pospisilik[3], Dustin E Schones[1,2]\***

[1]Department of Diabetes Complications and Metabolism, Beckman Research Institute, Duarte, United States; [2]Irell and Manella Graduate School of Biological Sciences, City of Hope, Duarte, United States; [3]Van Andel Research Institute, Grand Rapids, United States

**Abstract** Transposable elements (TEs) are mobile genetic elements that make up a large fraction of mammalian genomes. While select TEs have been co-opted in host genomes to have function, the majority of these elements are epigenetically silenced by DNA methylation in somatic cells. However, some TEs in mice, including the Intracisternal A-particle (IAP) subfamily of retrotransposons, have been shown to display interindividual variation in DNA methylation. Recent work has revealed that IAP sequence differences and strain-specific KRAB zinc finger proteins (KZFPs) may influence the methylation state of these IAPs. However, the mechanisms underlying the establishment and maintenance of interindividual variability in DNA methylation still remain unclear. Here, we report that sequence content and genomic context influence the likelihood that IAPs become variably methylated. IAPs that differ from consensus IAP sequences have altered KZFP recruitment that can lead to decreased KAP1 recruitment when in proximity of constitutively expressed genes. These variably methylated loci have a high CpG density, similar to CpG islands, and can be bound by ZF-CxxC proteins, providing a potential mechanism to maintain this permissive chromatin environment and protect from DNA methylation. These observations indicate that variably methylated IAPs escape silencing through both attenuation of KZFP binding and recognition by ZF-CxxC proteins to maintain a hypomethylated state.

**\*For correspondence:**
dschones@coh.org

## Introduction

Sequences derived from transposable elements (TEs) make up a large fraction of mammalian genomes (*Bourque et al., 2018*). TEs are silenced by both histone modifications and DNA methylation to prevent these elements from having deleterious impact on the host genome. Certain TEs, however, can escape silencing and behave as sites of interindividual epigenetic variability (*Bakshi et al., 2016*; *Dolinoy et al., 2007a*; *Gunasekara et al., 2019*; *Hernando-Herraez et al., 2015*; *Oey et al., 2015*; *Reiss et al., 2010*). These loci have been reported to be sensitive to environmental stimuli such as diet and exposure to toxins (*Dolinoy et al., 2007b*; *Morgan et al., 1999*; *Rosenfeld et al., 2013*; *Waterland et al., 2007*; *Waterland and Jirtle, 2003*), although this field remains an area of active research (*Bertozzi et al., 2021*). The well-most studied metastable epiallele is present in the Agouti viable yellow mouse, where genetically identical mice display phenotypic diversity that manifests as a range of coat colors and susceptibility to obesity (*Bernal et al., 2011*; *Rakyan et al., 2002*). This variable coat color has been linked to the methylation state of a novel insertion of an Intracisternal A-particle (IAP) retrotransposon (*Dolinoy et al., 2010*). Variable methylation of this IAP contributes to variable transcription of the *agouti* gene across all tissues and variation in coat color, hyperphagia and hyperinsulinemia (*Bernal et al., 2011*; *Rakyan et al., 2002*). These observations in the Agouti viable mouse model highlights the impact of variable methylation of TEs on phenotypic outputs. Thus, elucidating

the mechanisms underlying the establishment and maintenance of these variably methylated TEs is important to further our understanding of non-Mendelian phenotypic variability and epigenetic plasticity (*Ecker et al., 2018*; *Ecker et al., 2017*; *Tejedor and Fraga, 2017*).

Recent work profiling interindividual epigenetic variability in mice identified roughly 50 loci that are variably methylated in a manner similar to the IAP that drives the agouti mouse phenotype (*Adams et al., 2012*; *Bertozzi et al., 2020*; *Elmer et al., 2020*; *Kazachenka et al., 2018*). (*Adams et al., 2012*; *Kazachenka et al., 2018*). The majority of these variably methylated loci are IAP retrotransposons (VM-IAPs) (*Bertozzi et al., 2020*; *Faulk et al., 2013*; *Kazachenka et al., 2018*). IAPs are silenced in a sequence-specific manner by KRAB-domain containing zinc finger proteins (KZFPs) (*Coluccio et al., 2018*). KZFPs recruit KAP1, also known as TRIM28, which, in turn, recruits repressive protein complexes to deposit H3K9me3 and DNA methylation at these loci (*Bulut-Karslioglu et al., 2014*; *Ecco et al., 2017*; *Ecco et al., 2016*; *Schultz et al., 2002*; *Turelli et al., 2014*). While this has been thought to happen primarily in embryonic stem cells, recent work has demonstrated that KZFPs and KAP1 regulated TEs in somatic cells as well (*Ecco et al., 2016*). KZFPs have evolved in clusters through imperfect gene duplication and spontaneous mutations that can allow for the recognition of novel TEs (*Imbeault et al., 2017*; *Kauzlaric et al., 2017*). Over time, TEs can mutate and escape from KZFP targeted silencing (*Imbeault et al., 2017*; *Kauzlaric et al., 2017*; *Wolf et al., 2020*). It has been proposed that these KZFPs are engaged in an evolutionary arms race with TEs to maintain their silencing (*Jacobs et al., 2014*).

IAPs have an unusually high CpG content that is more similar to CpG islands (CGIs) than to the rest of the mouse genome (*Elmer et al., 2020*; *Kazachenka et al., 2018*). CGIs are generally hypomethylated and can recruit ZF-CxxC proteins, which help maintain a permissive chromatin environment (*Clouaire et al., 2012*; *Gu et al., 2018*; *Mikkelsen et al., 2007*; *Thomson et al., 2010*; *van de Lagemaat et al., 2018*). Notable CxxC domain containing proteins include CFP1, which is a subunit of the SET1 complex that deposit H3K4me3, and TET1/TET3, which are responsible for the active removal of methylation from CpG dinucleotides to maintain a hypomethylated state (*Gu et al., 2018*; *Tahiliani et al., 2009*; *Williams et al., 2011*). Mammalian genomes in general have an underrepresentation of CpG dinucleotides as a result of deamination at methylated cytosines over evolutionary time (*Duncan and Miller, 1980*; *Crayle et al., 2016*; *Shen et al., 1994*). The 'richness' of CpG dinucleotides at IAPs is potentially due to the fact these elements are more recent additions to the genome that have yet to undergo 'evolutionary' deamination at methylated CpG dinucleotides. It has been previously shown that recently evolved CpG dense TEs have the potential to be hypomethylated when there is no targeted suppression of these elements (*Jacobs et al., 2014*; *Long et al., 2016*; *Ward et al., 2013*).

Based on this evidence, we postulated that variably methylated CpG dense IAPs can have sequence variants that allow for partial escape from KZFP-mediated silencing (*Bertozzi et al., 2020*; *Faulk et al., 2013*; *Kazachenka et al., 2018*) and subsequently be bound by ZF-CxxC proteins, thereby preventing stable repression. Focusing on the IAPLTR1 subfamily identified as IAPLTR1_Mm in RepBase,, we performed multiple sequence alignment and hierarchical clustering of these elements and confirmed that the VM-IAPs have a unique sequence that is distinct from other IAPLTRs. These IAPLTRs have altered KZFP binding and diminished KAP1 recruitment. Relative to silenced IAPs – that are found in gene poor regions or proximal to tissue specific transcripts – we find that VM-IAPs are over-represented proximal to constitutively expressed genes. Importantly, we find that CpG dense TEs (regardless of subfamily) have increased recruitment of ZF-CxxC proteins such as CFP1 and TET1 in the absence of KZFP-mediated silencing.

## Results

### IAP sequence influences KZFP recruitment and establishment of variable methylation

Recent work has identified novel VM loci, which are largely IAPs of the IAPLTR1 and IAPLTR2 subfamilies, across tissues in C57BL/6 J mice (*Bertozzi et al., 2020*; *Elmer et al., 2020*; *Kazachenka et al., 2018*). IAPLTR1 and IAPLTR2 are among the most evolutionarily recent IAPs (*Qin et al., 2010*) As KZFPs are responsible for silencing IAP elements in a sequence-specific manner (*Coluccio et al., 2018*), we examined if the previously identified VM-IAPs have shared sequence variants that could allow for escape from KZFP-mediated silencing at both the LTR and internal element. While many

older IAPs have lost their internal elements over time, many IAPLTR1 and IAPLTR2, which among the most evolutionarily recent IAPLTRs, flank an IAPEz-int (*Qin et al., 2010*; *Stoye, 2001*). We performed multiple sequence alignment and hierarchical clustering of all LTRs belonging to the IAPLTR1 and IAPLTR2 subfamilies that were greater than 300 bps long using the ETE3 pipeline (*Huerta-Cepas et al., 2016*). This analysis treated all LTRs independently, regardless of whether the LTR was a solo LTR or part of an ERV. We also performed multiple sequence alignment and hierarchical clustering the first 150bps of IAPEz-ints flanked by either IAPLTR1 or IAPLTR2 elements, which has previously been show to recruit the KZFP Gm14419 (*Wolf et al., 2020*). This profiling resulted in the characterization of four sequence 'clades' for IAPLTR1s (*Figure 1A*, *Figure 1—figure supplements 1 and 2* and *Supplementary file 1*), two sequence 'clades' for the IAPLTR2s (*Figure 1—figure supplement 1* and *Supplementary file 1*), and three sequence clades for IAPEz-ints. All IAPLTRs flanking an IAPEz-int belonged to the same sequence clade.

To determine whether these clades have altered KZFP recruitment, we used published KZFP ChIP-seq profiles (*Wolf et al., 2020*) to survey the occupancy of KZFPs at each identified clade. We aligned all ChIP-seq libraries to the mm10 genome with multimapping, but allowed no SNPs, as we were specifically interested in identifying which sequences the KZFPs could bind (see Materials and methods for details). We identified that clade 1 IAPLTR1 elements have evidence of binding by ZFP989 and Gm21082, while clade 2 IAPLTR1 elements have evidence of binding by ZFP429 (*Figure 1A* and *Figure 1—figure supplement 3*). However, clade 3 and clade 4 IAPLTR1 elements do not show evidence of binding by these KZFPs and have decreased KAP1 binding (*Figure 1A and B* and *Figure 1—figure supplement 3*). Similar analysis with IAPLTR2s demonstrated that clade 1 IAPLTR2s are bound by ZFP429, while clade 2 IAPLTR2 elements are largely bound by ZFP989 (*Figure 1—figure supplement 4*). Both clades have evidence of KAP1 binding (*Figure 1—figure supplement 4*). When profiling KZFP and KAP1 occupation at the IAPEz-int variants, we observed that clade γ elements displayed the weakest Gm14419 binding and the lowest KAP1 signal compared to clades α and β (*Figure 1A and B*). Additionally, we observed that IAPLTR2s can flank either clade α or β IAPEz-ints, (*Figure 1—figure supplement 5*), while IAPLTR1s can flank any IAPEz-int, including IAPEz-int clade γ elements that have decreased KAP1 binding (*Figure 1C*). These results suggest that both LTR and internal sequence variants can result in altered KZFP recruitment and KAP1-mediated silencing.

Finally, we were interested in profiling whether the elements that have decreased KZFP-mediated silencing were significantly more likely to be variably methylated. We observed that VM-IAPLTR1s were significantly enriched at IAPs with IAPLTR1 clade three elements flanking IAPEz-int clade γ elements (*Figure 1C*), and VM-IAPLTR2 elements were significantly enriched at solo IAPLTR2 clade 1 elements (*P*-value < 1e-16 as determined using a Fisher exact test) (*Figure 1—figure supplement 5*). Consistent with previous work (*Bertozzi et al., 2020*; *Faulk et al., 2013*; *Kazachenka et al., 2018*), these results suggest that sequence is a contributing factor in the establishment of VM-IAPs. Overall, these results demonstrate that the IAPs that are most prone to be variably methylated are those that are escaping KZFP-mediated silencing.

## VM-IAP are proximal to constitutively expressed transcripts and enhancer elements

While sequence has been shown to be a factor in the establishment of variable methylation (*Bertozzi et al., 2020*; *Faulk et al., 2013*; *Kazachenka et al., 2018*), not all IAPs with sequences identical to VM-IAPs are variably methylated (*Figure 1A*), as has been reported previously (*Kazachenka et al., 2018*). Previous studies have identified that CTCF enrichment, a protein involved in regulating chromatin architecture, has been bound at these VM-IAPs when hypomethylated (*Elmer et al., 2020*; *Kazachenka et al., 2018*). However, as CTCF binding motifs appear to be present at both VM-IAPs and silenced IAPs (*Elmer et al., 2020*) we examined whether genomic context had an impact on the establishment of variable methylation at the VM-IAPs. We used proximity of the IAP to constitutively expressed genes (*Li et al., 2017*) and ENCODE annotated enhancer elements from any cell type (*Moore et al., 2020*) as our proxy for a euchromatic environment (see Materials and methods for details). We found that 83 % of the VM-IAPLTR1s were within 50 kb of a constitutively expressed gene, while the remaining 17 % of VM-IAPLTR1s were less than 1 kb from an annotated enhancer element. In contrast, only 12 % of the non-VM IAPs were proximal to a constitutively expressed gene and 7 %

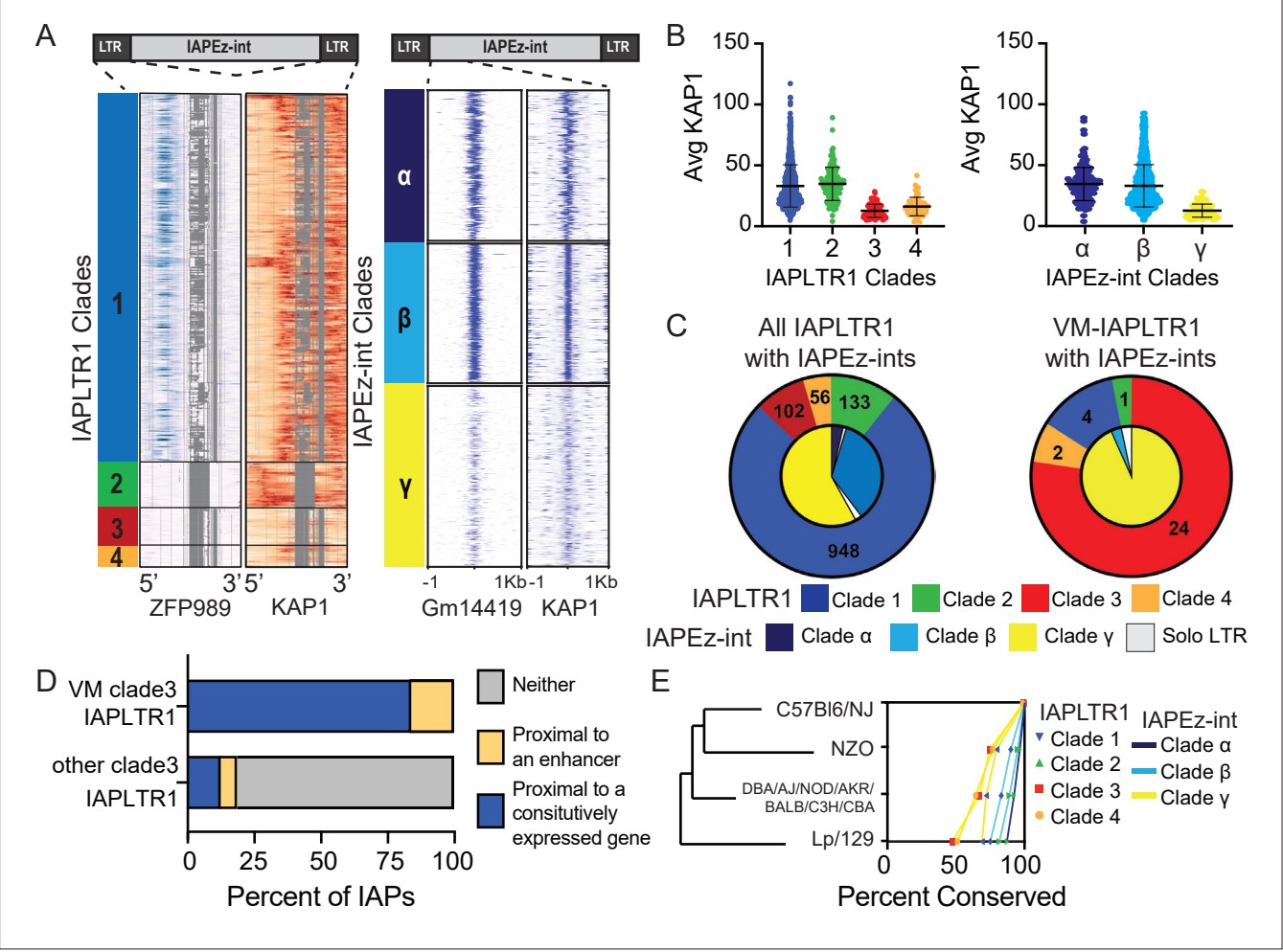

**Figure 1.** Sequence and chromatin context influence the establishment of a VM-IAPLTRs. (**A**) All IAPLTR1 elements larger than 300 bps and the first 150bps of IAPEz-ints flanked by IAPLTR1 or IAPLTRs were clustered by sequence using PhyML with default settings. Major sequence variants for IAPLTR1s were separated into separate clades: clade1 (blue), clade 2 (Green), clade 3 (red), and clade 4 (orange). Major sequence variants for IAPEz-ints were separated into separate clades: clade α (dark blue), clade β (light blue), and clade γ (yellow). KZFP and KAP1 ChIP-seq signal was mapped across the consensus IAPLTR1 sequence as determined by MAFFT multiple sequence alignment. Gaps in the multiple sequence alignment for the IAPLTR1 sequence are displayed as grey. Heatmaps for KZFP and KAP1 ChIP-seq signals centered on the 5' end of the IAPEz-int elements are show as well (**B**) Average KAP1 ChIP-seq signal across all IAPLTR1 and IAPEz-int clades. Each data point refers to the average KAP1 ChIP-seq signal at an individual IAPLTR1 element. Mean and standard deviation for each clade is shown as well. (**C**) Distribution of the IAPLTR1 elements clades (outer circle) and IAPEz-int clades (inner circle) for all IAP elements and VM-IAP elements. (**D**) Percent of VM-IAPLTR1s clade three elements and non VM-IAPLTR1 clade three elements that are within 50 kb of a constitutively expressed gene or 1 kb enhancer element, as a proxy for constitutive euchromatin environment. (**E**) Conservation of IAPLTR1 and IAPEz-int variants across mouse strains. Presence or absence of a IAP was determined using structural variants identified from the Sanger mouse genome project. Mouse KZFP and KAP1 ChIP-seq date are from GEO: GSE115291. VM-loci coordinates were obtained from *Elmer et al., 2020*.

The online version of this article includes the following figure supplement(s) for figure 1:

**Figure supplement 1.** Multiple sequence alignment of IAPLTR1s, IAPLTR2, and the first 150bps of IAPEz-int flanked by IAPLTR1/2 aligned by MAFFT with default settings (*Katoh and Standley, 2013*).

**Figure supplement 2.** Consensus sequence of the IAPLTR1 clades identified in *Figure 1—figure supplement 1*.

**Figure supplement 3.** Gm21082 and KAP1 ChIP-seq signal mapped across all IAPLTR1 elements.

**Figure supplement 4.** ZFP429, ZFP989, and KAP1 ChIP-seq signal mapped across all IAPLTR2 elements.

**Figure supplement 5.** Association of IAPLTR2 and IAPEz-int clades for all IAPLTR2s and VM-IAPLTR2.

**Figure supplement 6.** The percent of VM-IAPLTR1s clade three elements and non VM-IAPLTR1 clade three elements proximal to a constitutively expressed genes.

*Figure 1 continued on next page*

*Figure 1 continued*

**Figure supplement 7.** Conservation of IAPLTR1 and IAPEz-int variants across mouse strains.

**Figure supplement 8.** Model of IAPLTR and IAPEz-int interactions the height of each IAPEz-int refers to the percentage of IAP elements.

were proximal to an enhancer (*Figure 1D*). Clade 3 VM-IAPLTR1s were more likely to be proximal to a constitutively expressed gene than other clade 3 IAPLTR1s regardless of the FPKM or distance profiled (*Figure 1—figure supplement 6*). This indicates that while sequence allows these elements to have the potential to become variably methylated, VM-IAPs are present in a permissive chromatin environment to escape being silenced, as heterochromatin spreading from neighboring loci may drive silencing of these elements.

## Conservation of IAPs between mouse strains

IAP elements are a potential source of novelty between mouse strains and VM-IAPs in particular have been shown to be highly polymorphic (*Gagnier et al., 2019*; *Kazachenka et al., 2018*; *Nellåker et al., 2012*; *Rebollo et al., 2020*). In order to investigate the relationship between IAP sequence, potential for variable methylation and conservation across strains, we leveraged structural variant differences between mouse strains identified by the Sanger mouse genome project (*Keane et al., 2011*). We found that clade 3–4 IAPLTR1s, which have the lowest KAP1 binding of the IAPLTR1 clades and IAPEz-int clade γ elements are more polymorphic across mouse strains than the other clades (*Figure 1E*). Additionally, we found that IAPLTR2 clade 1 elements, which are more prone to be variably methylated, are more likely to be polymorphic across mouse strains (*Figure 1—figure supplement 7*). Thus, IAPs that have diminished KZFP recruitment are more likely to be polymorphic between mouse strains (*Figure 1—figure supplement 8*).

## IAPs that have loss of KZFP binding are bound by the ZF-CxxC containing proteins TET1 and CFP1

It has previously been reported that IAPLTR1s and IAPLTR2s have a significantly higher CpG density than the genomic average (*Elmer et al., 2020*; *Kazachenka et al., 2018*). Given the prevalence of variable methylation at these elements, we wanted to assess the relationship between CpG content and variable methylation directly. For each TE subfamily, we profiled the CpG density of all LTRs (see Materials and methods for details) and the percentage of elements that were variably methylated using the previously identified list of VM-TEs in C57BL/6 J mice (*Kazachenka et al., 2018*). We observed that TE subfamilies with a higher average CpG density had a larger percentage of elements that are variably methylated, with IAPLTRs being the most CpG dense and having the largest percentage of variably methylated elements (*Figure 2A*). The CpG content of VM-IAPLTR1s and VM-IAPLTR2s is similar to non-VM-IAPLTR1s and VM-IAPLTR2s, indicating that high CpG content alone is not sufficient to induce variable methylation (*Figure 2B*). As CpG dense loci are capable of being recognized and bound by ZF-CxxC proteins (*Blackledge et al., 2010*; *Thomson et al., 2010*), we examined the ability of ZF-CxxC proteins to bind to IAPLTR1s. Utilizing a publicly available ChIP-seq dataset for the ZF-CxxC-domain containing protein TET1 in mouse embryonic stem cells (mESCs) (*Gu et al., 2018*), we identified increased TET1 binding at clade3 IAPLTR1s elements, which contains the most variably methylated elements (*Figure 2C* and *Figure 2—figure supplement 1*). We furthermore profiled CpG methylation of VM-IAPLTRs in *Tet1* knockout (KO) and *Tet1/Dnmt3a* double knockout (DKO) mESCs (*Gu et al., 2018*) and found that VM-IAPs displayed a significant increase in CpG methylation in the *Tet1* KO cells (*Figure 2D*). This increase in methylation is reversed when both *Dnmt3a* and *Tet1* are knocked out in mESCs (*Figure 2D*). We also profiled the recruitment of another ZF-CxxC protein, CFP1, using existing CFP1 ChIP-seq data from C57Bl/6 mice and observed CFP1 binding appears to be specific to VM-IAPLTRs (*Figure 2—figure supplement 2*). As CFP1 is a subunit of the SET1 complex responsible for depositing H3K4me3, we examined publicly available H3K4me3 ChIP-seq data from B cells (*Adams et al., 2012*) to determine if H3K4me3 is enriched at VM-IAPs. Consistent with previous observations (*Bertozzi et al., 2020*), we observed that VM-IAPs were more likely to ben enriched for H3K4me3 compared to non-variable IAPs (*Figure 2—figure supplement 3*). De novo motif discovery applied to VM-IAP sequences furthermore revealed known recognition sequences for ZF-CxxC proteins such as CFP1 (*Figure 2—figure supplement 4*). ZF-CxxC protein binding has been

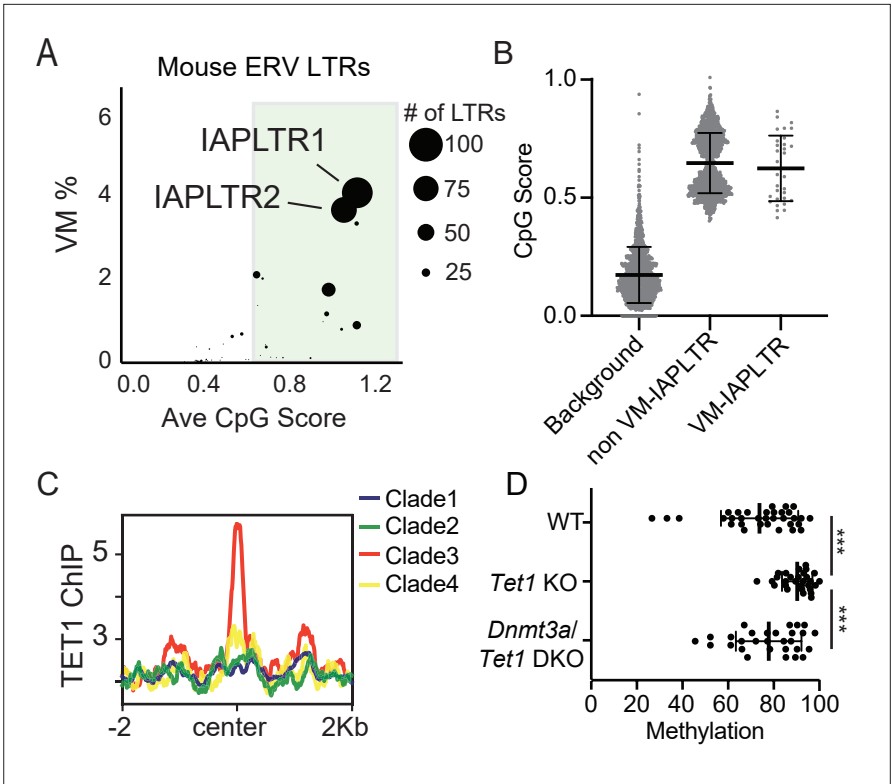

**Figure 2.** Divergent VM-IAPs elements have high CpG density and recruitment of ZF-CxxC proteins. (**A**) Percent of ERV LTRs elements that are variably methylated for a given TE subfamily and average maximum CpG score of the TE subfamily. The size of each dot is determined by the number of VM-loci for each subfamily, with the largest dots indicating the greatest number of variably methylated elements. Average CpG score was determined by identifying the most CpG dense 200 bp window of each LTR and calculating the average CpG score for the whole subfamily. (**B**) Average CpG density of silenced and variably methylated IAPLTRs, as well as the average CpG score of a randomly selected background the same size as the IAPLTRs. Each dot refers to an individual IAPLTR element in the mm10 genome. Mean and standard deviation for each group is shown. (**C**) Aggregate plots of TET1 ChIP-seq signal across all identified IAPLTR1 clades. (**D**) Average methylation percentage of non-variable and VM-IAPLTR elements in wild type, Tet1 knockout (KO), and Dnmt3a/Tet1 double knockout cells (DKO). Each dot refers to the average methylation of an individual IAPLTR element. Only CpGs with >5 x coverage were retained to calculate methylation. Significance determined using a Wilcoxon rank sum test. *** indicates a p.value <0.0001. Bars for mean and standard deviation is shown as well for (**B and D**). TET1 ChIP-seq from GEO:GSE100957. Mouse *Tet1* and *Tet1/Dnmt3a* DKO WGBS from GEO:GSE134396.

The online version of this article includes the following figure supplement(s) for figure 2:

**Figure supplement 1.** TET1 ChIP-seq signal mapped across all IAPLTR1 elements.

**Figure supplement 2.** CFP1 profiling at IAPLTRs.

**Figure supplement 3.** Enrichment of H3K4me3 at VM-IAPLTRs.

**Figure supplement 4.** MEME identified motifs present in VM-IAPLTRs that contain CFP1-binding sites.

shown to protect the underlying DNA from methylation (*Blackledge et al., 2010*; *Thomson et al., 2010*). From these results we propose that recruitment of ZF-CxxC proteins, which have been shown to be involved in maintaining a euchromatic state at CpG islands, may be involved in the establishment of variable methylation at VM-IAPs in mice.

To determine if evolutionarily recent CpG-dense TEs are also more prone to be variably methylated in humans, we used a previously identified list of loci that display interindividual epigenetic variation in humans (*Gunasekara et al., 2019*). We determined the observed frequency at which variably methylated loci were found to be present on a TEs and compared this to an expected distribution if

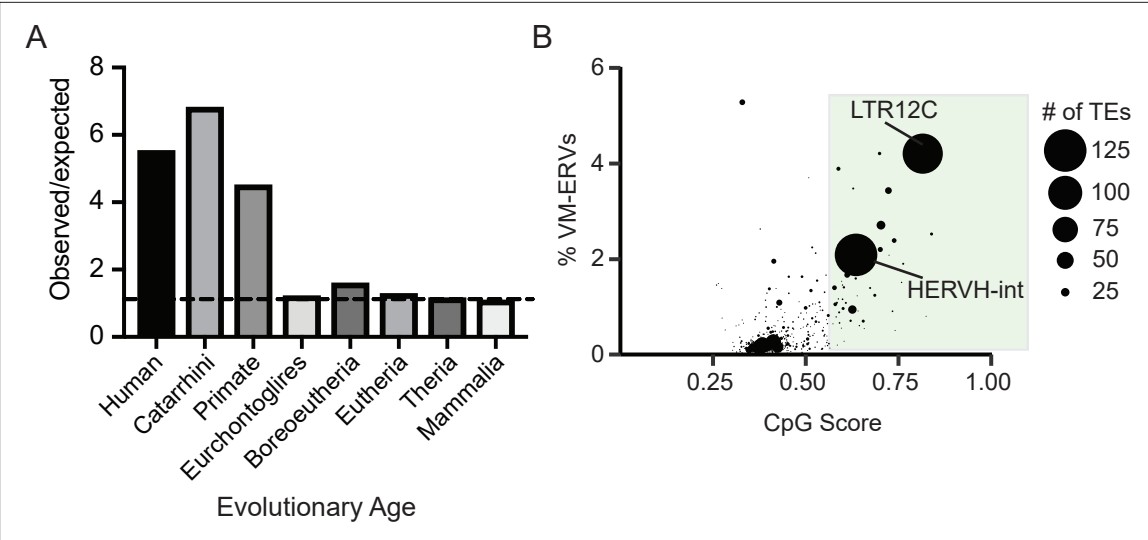

**Figure 3.** Variably methylated loci in humans are enriched for evolutionarily recent CpG-dense TEs. (**A**) Observed over expected distribution of VM-TE elements in humans stratified by the evolutionary age of the TE. Expected distribution was determined using a random sampling of the hg38 genome the same size as the VM-loci. Evolutionary age for each TE subfamily was obtained from DFAM. (**B**) Scatterplot showing the percent of LTR elements which display VM for a given TE subfamily and average CpG score of the TE subfamily. The size of each dot is determined by the number of VM-loci for each subfamily. Average CpG score was determined by identifying the most CpG dense 200 bp window of each ERV and calculating the average CpG score for the whole subfamily. The coordinates of VM-loci for humans were obtained from *Gunasekara et al., 2019*.

The online version of this article includes the following figure supplement(s) for figure 3:

**Figure supplement 1.** Scatter plot showing the percent of SINEs and LINEs that display variable methylation for a given TE subfamily and average CpG score of the TE subfamily in humans.

the variable methylation was only due to chance. Expected distribution was determined by selecting random loci in the genome the same size of the variably methylated loci. We observed that evolutionarily recent TEs were overrepresented in the variably methylated dataset (*Figure 3A*). TE subfamily ages were obtained from DFAM (*Hubley et al., 2016*). We also found the TE subfamilies with a higher CpG density are more likely to be variably methylated (*Figure 3B* and *Figure 3—figure supplement 1*), with the highly CpG dense transposable element subfamily LTR12C having both one of the highest CpG densities and one of highest percentage of elements that are variably methylated. LTR12Cs are recently evolved TEs that have previously been shown to have latent regulatory potential (*Ward et al., 2013*) and can act as cryptic promoter elements in cancer cells (*Babaian and Mager, 2016*). These results demonstrate a correlation between CpG density and variable methylation in both mice and humans.

## CpG dense TEs are hypomethylated and recruit ZF-CxxC proteins in the absence of KZFP-mediated silencing

To further investigate the relationship between KZFP/KAP1 silencing, DNA methylation, and ZF-CxxC binding to CpG dense TEs, we profiled DNA methylation and ZF-CxxC binding in Tc1 mouse liver. This 'transchromosomic' mouse carries the majority of human chromosome 21 (Hsa21), which possesses human specific transposable elements without the corresponding KZFPs for silencing (*O'Doherty et al., 2005*) and has previously been used to identify the latent regulatory potential of primate specific TEs (*Jacobs et al., 2014*; *Long et al., 2016*; *Ward et al., 2013*). We performed MinION nanopore sequencing to globally profile DNA methylation of human chromosome 21 in Tc1 mouse liver and compared this with existing DNA methylation data from human liver cells (*Li et al., 2016*; *Figure 4A and B*). Consistent with previous reports (*Jacobs et al., 2014*; *Long et al., 2016*; *Ward et al., 2013*), we found that TE-derived CGIs were hypomethylated in the absence of KZFP-mediated silencing (*Figure 4B*). In contrast, TE-derived CGIs that can be targeted by KZFPs shared between mice and humans were largely methylated (*Figure 4—figure supplement 1*). These silenced TE-derived CGIs are largely Alu elements that are derived from the same ancestral sequence as mouse B2

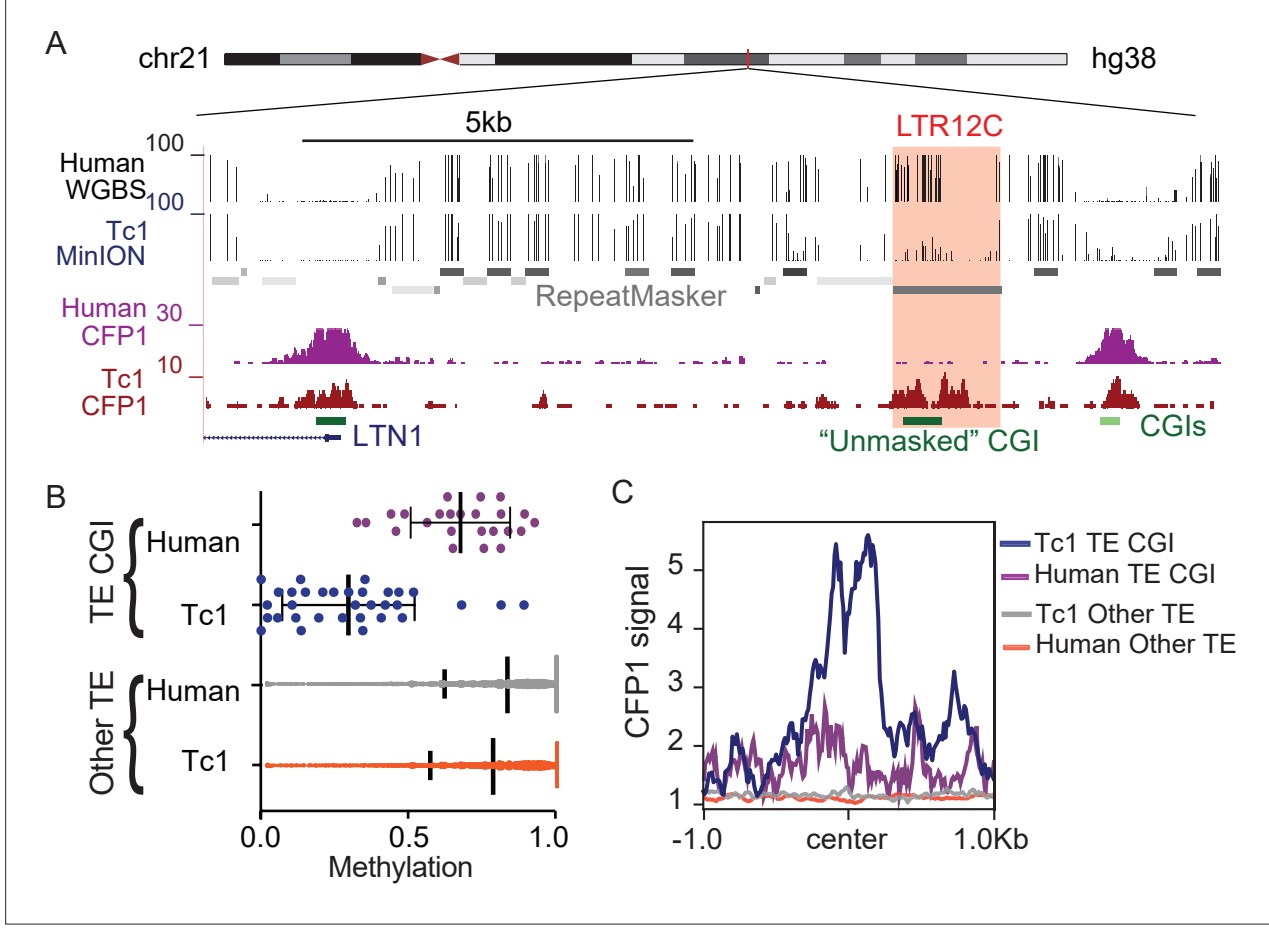

**Figure 4.** CpG dense TEs are hypomethylated and recruit ZF-CxxC proteins in the absence of KZFP-mediated silencing. (**A**) UCSC genome browser screenshot of an LTR12C element which is hypomethylated in Tc1 mice and shows novel CFP1 recruitment in Tc1 mice. (**B**) CpG methylation of TE CpG islands (CGIs) and other TEs for human chromosome 21 in both Tc1 mouse and human genomes. Each data point refers to an individual CpG island that had greater than 5 x coverage. Mean and standard deviation for DNA methylation at is shown as well. (**C**) CFP1 signal in Tc1 mice and humans at the human TE CGIs and other TEs (from **B**) on human chromosome 21. Human CFP1 ChIP-seq from GEO:GSM3132538. WGBS from human liver GEO:GSM1716957.

The online version of this article includes the following figure supplement(s) for figure 4:

**Figure supplement 1.** CpG methylation of all TEs on human chromosome 21.

**Figure supplement 2.** CFP1 signal at all non-repeat derived CpG islands on human chromosome 21 in Tc1 mice and humans.

SINE elements and can be silenced by the KZFP ZNF182, which is present in both mice and humans (*Imbeault et al., 2017*). Additionally, other primate specific TEs that are not CpG rich are methylated in both the Tc1 and human livers (Figured 4B), highlighting the importance of CpG density in escaping CpG methylation at TEs. To profile ZF-CxxC occupancy at these hypomethylated TEs, we performed CUT&RUN for CFP1, a ZF-CxxC protein expressed in somatic tissue, in Tc1 mouse livers and compared this with existing CFP1 ChIP-seq data from human erythroblasts (*van de Lagemaat et al., 2018*). We observed that CpG dense TEs that are not targeted by KZFPs displayed CFP1 recruitment, while the methylated CpG poor TEs have no CFP1 signal (*Figure 4C*). This CFP1 binding was also unique to the Tc1 mice as there was no CFP1 binding at these loci in humans (*Figure 4C*), while non-TE CGIs in both the mouse and human genomes have CFP1 binding (*Figure 4—figure supplement 2*). Together this work demonstrates that in the absence of targeted KZFP-mediated silencing, TEs with high CpG content can be hypomethylated and bound by ZF-CxxC proteins.

## Trim28 haploinsufficiency activates evolutionarily recent TEs

As TRIM28/KAP1 is a key protein involved in the establishment of heterochromatin at TEs, we profiled *Trim28* haploinsufficient mice to determine if decreased abundance of TRIM28 allows for CpG dense TEs to escape silencing. *Trim28* haploinsufficiency has previously been shown to impact the Agouti viable yellow mouse coat color (*Daxinger et al., 2016*) and drive a bimodal obesity phenotype in

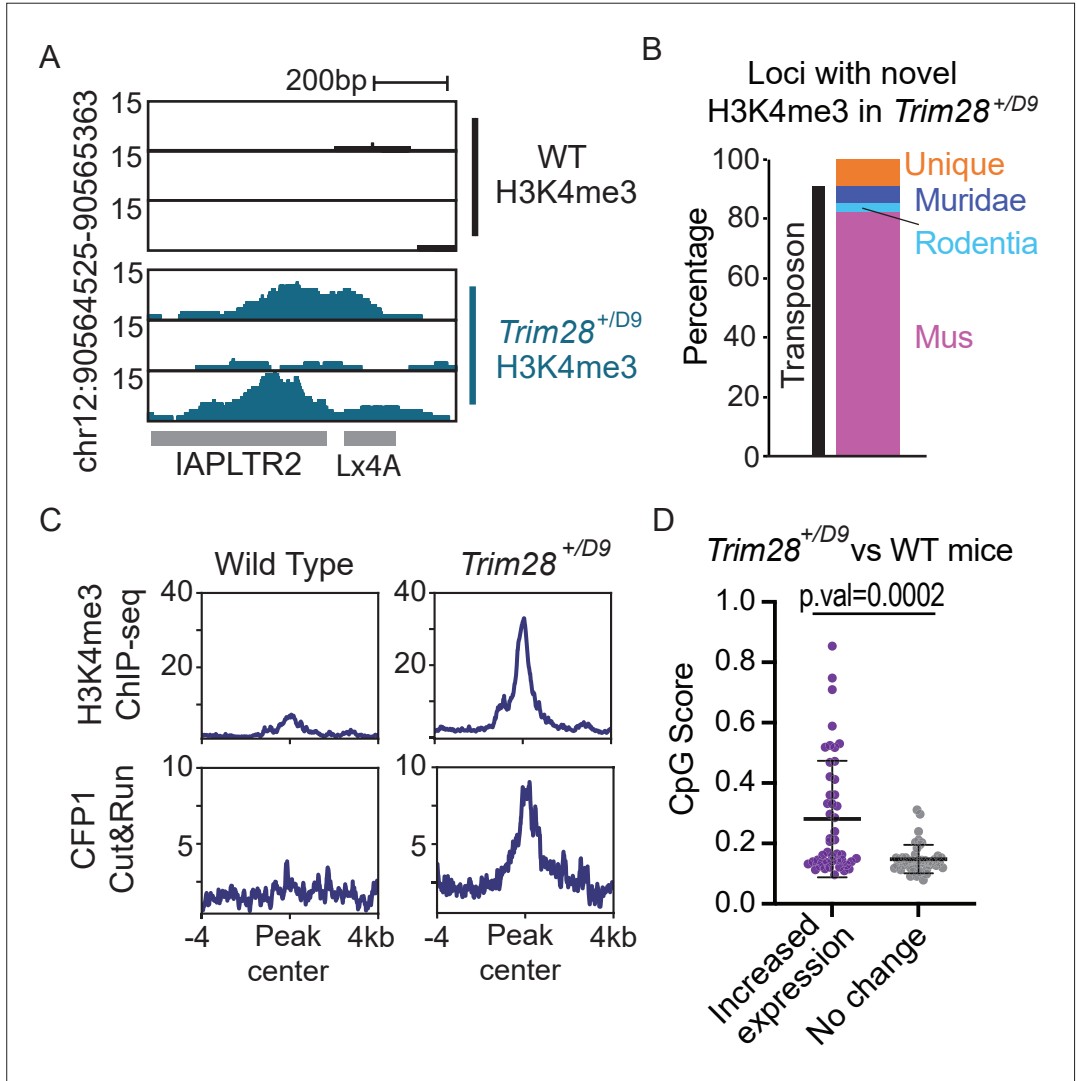

**Figure 5.** Trim28 haploinsufficiency leads to activation of evolutionarily recent and CpG dense TEs. (**A**) Genome screenshot of an IAPLTR2 element with novel H3K4me3 enrichment in Trim28 haploinsufficient mice. (**B**) Breakdown of loci with novel H3K4me3 in Trim28 haploinsufficient mice. Age of each TE was determined by DFAM. (**C**) Aggregate plots of H3K4me3 and CFP11 signal across all loci that have novel H3K4me3 signal in Trim28 haploinsufficient mice (**D**) CpG Score of TE subfamilies with a global increase expression in Trim28 haploinsufficient mice and a random selection of non-responsive TEs subfamilies. Expression levels for each TE subfamily was determined using RepEnrich, and Deseq2 was used to determine TE subfamilies with a significant increase in expression in the Trim28 haploinsufficient mice. Bar is placed at mean and error bars cover one standard deviation. Each data point refers to an individual TE subfamily. p-Values for CpG density difference was calculated using a Wilcoxon rank sum test. *Trim28* haploinsufficient and wild-type mouse RNA-seq was obtained from ENA:PRJEB11740.

The online version of this article includes the following figure supplement(s) for figure 5:

**Figure supplement 1.** Heatmap and aggregate plots of H3K4me3 and CFP1 signal at loci with novel H3K4me3 signal in Trim28 D9/+ mice.

**Figure supplement 2.** Evolutionary age of all TEs in the mm10 genome.

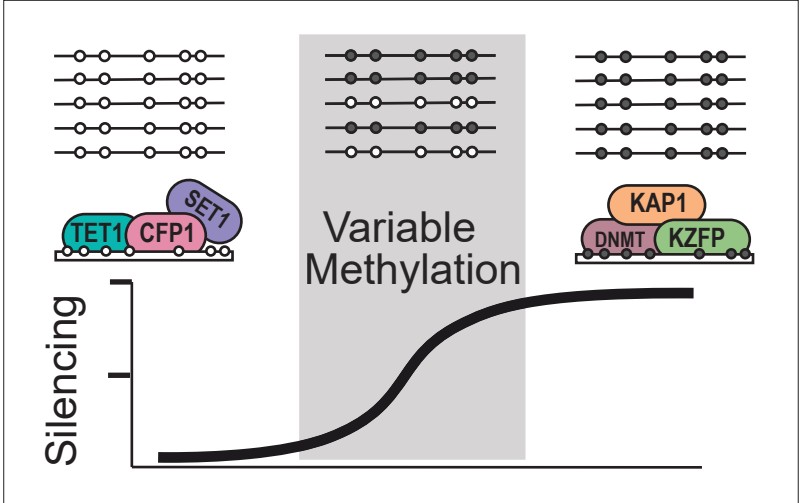

**Figure 6.** Model for variable methylated transposable elements. Loci with high CpG density and loss of KZFP binding have the potential to recruit ZF-CxxC proteins to protect these TEs from being silenced. However, elements with high CpG density but strong KZFP recruitment will remain methylated.

The online version of this article includes the following figure supplement(s) for figure 6:

**Figure supplement 1.** Comparison of the IAPLTR1 associated with the 'master' IAP element identified in the C3H mice compared to the consensus sequence for each clade.

FVB/NJ mice (*Dalgaard et al., 2016*). We performed both H3K4me3 ChIP-seq and CFP1 CUT&RUN in the livers of Trim28 haploinsufficient (*Trim28$^{+/D9}$*) mice and their wild-type littermates (Materials and methods; *Figure 5A*). We identified 69 loci that had novel H3K4me3 signal in the *Trim28$^{+/D9}$* mice compared to control (Materials and methods; *Figure 6—figure supplement 1*). The majority of these loci with novel H3K4me3 recruitment were evolutionarily recent TEs (*Figure 6B and* - supplement 2). These loci also had an increase in CFP1 signal in *Trim28$^{+/D9}$* mice relative to control mice (*Figure 6C* and *Figure 6—figure supplement 1*). Additionally, we profiled global changes in expression of TE subfamilies in the *Trim28$^{+/D9}$* mice compared to wild-type mice using our previously generated RNA-seq datasets (*Dalgaard et al., 2016*). We identified that TEs which have a significant increase in gene expression in the *Trim28$^{+/D9}$* mice compared to wild-type mice have significantly higher CpG density than expected when compared to non-responsive TE subfamilies (*Figure 6D*). Together this work demonstrates that loss of KAP1-mediated silencing allows for some evolutionary recent TEs to be active and recruit ZF-CxxC proteins.

## Discussion

This work provides the foundation for a deeper understanding of how mammalian genomes are shaped during evolution. It has been proposed that genomes are locked in an 'evolutionary arms race' between endogenous retrovirus and KZFPs to maintain proper silencing of the genome. The variable methylation observed at IAPs could be the result of the IAPs winning this 'battle'. We observed that VM-IAPs have unique sequence variants that allow them to have diminished KZFP recruitment. This provides an explanation to previous works that identified sequence biases for variably methylated IAPs in mice (*Bertozzi et al., 2020*; *Faulk et al., 2013*; *Kazachenka et al., 2018*). We also found that IAPs with the same sequence variants as the VM-IAPs are more likely to be polymorphic between individual mouse strains. Interestingly, we find that the IAPLTRs flanking the 'master' IAP that has been proposed to be responsible for an expansion of IAPs in C3H/HeJ mice (*Rebollo et al., 2020*) has the same sequence as the clade 3 IAPLTR1s (*Figure 6—figure supplement 1*). Together this implies that these VM-IAPs may be more active in certain mouse strains and may be propagating due to a lack of repression.

Our work also highlights the potential importance of CpG density in the silencing of the endogenous retroviruses. We observed that VM-IAPs have recruitment of ZF-CxxC proteins. In the absence

of targeted silencing, CpG-dense TEs have the potential to become hypomethylated and recruit ZF-CxxC proteins, while CpG poor TEs remain methylated. As TEs can lose CpG density over time, the observation that the CpG rich TEs have the potential for reactivation supports the idea that deamination of methylated loci could be one means for driving permanent silencing of TEs over time (*Long et al., 2016*).

Based on our results, we propose a model that VM-IAP loci are established as a result of partial KZFP mediated silencing at CpG dense TEs, which can then be protected from silencing through recruitment of ZF-CxxC containing proteins such as TET1 (*Figure 6*). This potentially results in a stochastically silenced locus, which is then inherited by all tissues during development. Together our work has significant implications in furthering our understanding of how TEs can alter both the genomes and epigenomes of mammals.

# Materials and methods

**Key resources table**

| Reagent type (species) or resource | Designation | Source or reference | Identifiers | Additional information |
|---|---|---|---|---|
| Strain, strain background (*Mus musculus*) | *Trim29+/D9* | *Blewitt et al., 2005* | (RRID:MGI:3821610) | Haploinsufficient for *Trim28* |
| Strain, strain background (*Mus musculus*) | B6129S-Tc(HSA21)1TybEmcf/J | The Jackson Laboratory | Stock No: **010801** (JAX) (RRID:IMSR_JAX:010801) | 2 Mb of a freely segregating human fragment of Chr21 |
| Antibody | Rabbit Polyclonal anti-CFP1 antibody | Millipore | ABE211 (RRID:AB_10806210) | CUT&RUN (1:50 dilution) |
| Antibody | Rabbit polyclonal anti-H3K4me3 Antibody | Abcam | ab8580 (RRID:AB_306649) | ChIP-seq (2 µg antibody per 25 µg chromatin) |

## Multiple sequence alignment and hierarchical clustering

IAPLTR1_Mm, IAPLTR2_Mm, and IAPEz-int elements were defined by RepeatMasker (RRID:SCR_012954) (Smit, AFA, Hubley, R & *Smit et al., 2010*) in the mm10 genome. Only IAPLTRs greater than 300bps in length were profiled. For IAPEz-ints, only IAPEz-int elements flanked by either IAPLTR1 or IAPLTR2 elements were selected for profiling, and only the first 150bps of the 5' end of the IAPEz-int were aligned and clustered. All elements were processed using the ETE3 pipeline (*Huerta-Cepas et al., 2016*) using MAFFT (RRID:SCR_011811) with default settings (*Katoh and Standley, 2013*), and bases that were found in less than 10 % of the samples were trimmed from the multiple sequence alignment using trimAL (RRID:SCR_017334) (*Capella-Gutiérrez et al., 2009*). The multiple sequence alignments were then hierarchically clustered using PhyML (RRID:SCR_014629) (*Guindon et al., 2010*). Sequence clades were empirically selected.

## Alignment of existing ChIP-Seq data

Reads were trimmed using Trimgalore version 0.5.0 (RRID:SCR_011847), which utilizes cutadapt (RRID:SCR_011841) (*Martin, 2011*), and were aligned to either the mm10 genome for mouse data, hg38 genome for human data, or a custom assembly that included both the mm10 genome and human chromosome 21 for the Tc1 mice. Reads were aligned using bowtie1 version 1.2.3 (RRID:SCR_005476) retaining only reads that could be mapped to unambiguously to a single locus using the -m one option (*Langmead et al., 2009*). Aligned reads were sorted using samtools version 1.10 (RRID:SCR_002105) (*Li et al., 2009*) and filtered to remove duplicate reads using the MarkDuplicates function of Picardtools version 2.21.1 (RRID:SCR_006525). Regions of enrichment were called using the callpeaks function of MACS2 version 2.2.5 (RRID:SCR_01329) (*Zhang et al., 2008*) with a q-val threshold of 1e-3.

KZFP and KAP1 data in *Figure 1* was aligned using different parameters to allow for potential multimapping as the reads were too short to provide confident unique mapping at repetitive elements. These reads were aligned to the mm10 genome using bowtie2 version 2.3.5.1 (RRID:SCR_016368) with the `--end-to-end --very-sensitive` options (*Langmead and Salzberg, 2012*). Aligned reads were sorted using samtools version 1.10 (RRID:SCR_002105) (*Li et al., 2009*) and filtered for duplicate reads using the MarkDuplicates function of Picardtools version 2.21.1 (RRID:SCR_006525).

Regions of enrichment were called using MACS2 version 2.2.5 callpeaks (RRID:SCR_01329) (*Zhang et al., 2008*) with a q-val threshold of 1e-3.

For all datasets, Bedgraph files were generated using bedtools version 2.29.0 genomecov (RRID:SCR_006646) with the -bg -ibam options (*Quinlan and Hall, 2010*). BigWigs (RRID:SCR_007708) were generated using the UCSCtools bedGraphToBigWig (*Karolchik et al., 2004*). Heatmaps of ChIP-seq signal across the IAP multiple sequence alignments were generated using a custom script to profile the read coverage at each base and were visualized using pheatmap (RRID:SCR_016418). All other heatmaps and aggregate plots of loci that extend were generated using deeptools (RRID:SCR_016366) (*Ramírez et al., 2016*).

### Genomic context analysis

We used proximity to a constitutively expressed gene or annotated enhancer element as a proxy for profiling a euchromatic environment. Constitutive expressed genes were identified as genes that have a fragments per kilobase of transcript per million mapped reads (FPKM) greater than two in more than 90 % of the samples previously profiled in *Li et al., 2017*, while enhancers were all ENCODE annotated enhancer elements (RRID:SCR_006793) (*Moore et al., 2020*). While FPKM of >2 was used, observed trends appeared regardless of FPKM threshold set (*Figure 1—figure supplement 6*).

### Polymorphism analysis between mouse strains

Coordinates of the structural variants between mouse strains were obtained from the Sanger mouse genome project (RRID:SCR_006239) (*Keane et al., 2011*). Deletions between mouse strains were extracted for each of the profiled mouse strains. Bedtools version 2.29.0 intersect (RRID:SCR_006646) (*Quinlan and Hall, 2010*) with the -f one option was used to determine if an IAP was fully deleted in each mouse strain relative to C57BL/6 J. Phylogeny of the mouse strains was obtained from previous work (*Nellåker et al., 2012*). For branches where multiple mouse strains are similarly diverged from C57BL6/J, the IAP was considered to be present if the IAP existed in any of profiled mice at that branch.

### Profiling of CpG score

CpG score was calculated as previously described (*Gardiner-Garden and Frommer, 1987*). Briefly, we calculated CpG score as Obs/Exp CpG = Number of CpG * N / (Number of C * Number of G). When profiling the CpG density of across the TE using sliding 200 bp tiles with 10 bp steps between windows and selected the most CpG dense portion of the TE to remove and impact of TE size. Window size was chosen to match the minimum size of a CpG island (*Gardiner-Garden and Frommer, 1987*). Differences in CpG density across an element was most pronounced on L1 elements, which can contain a CpG island in their 5' UTR but have low CpG density across the rest of the element. TEs with low copy numbers, such as IAPLTR4, were removed from this analysis.

### MinION sequencing and alignment

Livers from B6129S-Tc(HSA21)1TybEmcf/J mice (RRID:IMSR_JAX:010801) were purchased from JAX. Purified DNA from Tc1 mouse liver was prepared using the SQK-LSK109 ligation sequencing kit and run on R9 flow cells purchased from Nanopore. 1 ug of DNA was end repaired with NEBNext FFPE DNA repair and Ultra II End-prep purchased from New England Biolabs. The repaired DNA was cleaned with KAPA Pure Beads. Adapters were ligated using NEBNext quick T4 ligase purchased from New England Biolabs. Prepared libraries were mixed with Sequencing Buffer from the SQK-LSK109 kit and approximately 200 ng was loaded onto the MinION flowcell. Reads were aligned using bwa mem with the options -x ont2d -t 100 (RRID:SCR_010910) (*Li, 2013*) to a custom assembly that included all of the mm10 genome and human chromosome 21, and then sorted using samtools version 1.10 (RRID:SCR_002105) (*Li et al., 2009*). Reads were processed using minimap2 (RRID:SCR_018550) with the options -a -x map-ont (*Li, 2018*). Methylation state of CpGs was called using nanopolish (RRID:SCR_016157) with the options call-methylation -t 100 (*Loman et al., 2015*). Only loci with greater than 5 x coverage were considered in the analysis. CpG islands were obtained from the UCSC table browser (*Gardiner-Garden and Frommer, 1987*; *Karolchik et al., 2004*). Methylation percentage was averaged across CpG islands.

## CFP1 CUT&RUN sequencing and alignment

CUT&RUN was performed as previously described (*Skene and Henikoff, 2017*) on livers from B6129S-Tc(HSA21)1TybEmcf/J mice (RRID:IMSR_JAX:010801) purchased from JAX and Trim28$^{+/D9}$ livers using the CFP1 antibody (RRID:AB_10806210) (ABE211, Millipore). Briefly, unfixed permeabilized cells are incubated with the CFP1 antibody fused to A-Micrococcal Nuclease. Fragmented DNA was isolated and sequenced on an Illumina HiSeq 2,500 System. We assessed standard QC measures on the FASTQ file using FASTQC (RRID:SCR_014583) and adapters were trimmed using Trimgalore version 0.5.0 (RRID:SCR_011847). Trimmed reads were aligned to a custom assembly that included both the mm10 genome and human chromosome 21, or only the mm10 genome using bowtie1 version 1.2.3 (RRID:SCR_005476) using the -m one option to only retain reads that could be mapped to unambiguously to a single locus (*Langmead et al., 2009*). Aligned reads were sorted using samtools version 1.10 (RRID:SCR_002105) (*Li et al., 2009*) and filtered for duplicate reads using the MarkDuplicates function of Picardtools version 2.21.1 (RRID:SCR_006525). Regions of CFP1 enrichment were called using the call peaks function of MACS2 version 2.2.5 (RRID:SCR_01329) (*Zhang et al., 2008*) with a *q*-val threshold of 1e-3. Only peaks found in two animals were retained to remove biological noise. Additionally, due to small fragment size, loci that could not be mapped unambiguously by 90 bp fragments were removed from consideration using the deadzones tool from RSEG (RRID:SCR_007695) (*Song and Smith, 2011*).

## Animals

The generation of *Trim28$^{+/D9}$* mice (RRID:MGI:3821610) has been described elsewhere (*Blewitt et al., 2005*). Animals were kept on a 12 hr light/dark cycle with free access to food and water and housed in accordance with international guidelines. Livers were isolated at 2 weeks of age and immediately snap-frozen for further processing.

Tc1 mouse livers (RRID:IMSR_JAX:010801), freshly harvested and snap frozen, were purchased from JAX Laboratories (Stock No: 010801).

## ChIP-Seq analysis of H3K4me3 ChIP-Seq

Chromatin immunoprecipitation was performed with an H3K4me3 antibody (RRID:AB_306649) (ab8580, Abcam) as previously described (*Leung et al., 2013*). Isolated DNA was sequenced on an Illumina HiSeq 2,500 System. We assessed standard QC measures on the FASTQ file using FASTQC (RRID:SCR_014583) and adapters were trimmed using Trimgalore version 0.5.0 (RRID:SCR_011847). Reads were aligned to the mm10 genome using bowtie1 version 1.2.3 (RRID:SCR_005476) using the -m one option to only retain reads that could be mapped to unambiguously to a single locus (*Langmead et al., 2009*). Aligned reads were sorted using samtools version 1.10 (RRID:SCR_002105) (*Li et al., 2009*) and filtered for duplicate reads using the MarkDuplicates function of Picardtools version 2.21.1 (RRID:SCR_006525). Regions of enrichment were called using the call peaks function of MACS2 version 2.2.5 (RRID:SCR_01329) (*Zhang et al., 2008*) with a q-val threshold of 1e-3. Loci that displayed enrichment of H3K4me3 in at least two animals were retained to remove biological noise. The loci with increased H3K4me3 as shown in *Figure 6B–C* were identified by finding peaks present in *Trim28$^{+/D9}$* mice that were absent from WT mice and also had 3 x more H3K4me3 signal in *Trim28$^{+/D9}$* mice compared to WT. Heatmaps and aggregate plots were generated using deeptools (RRID:SCR_016366) (*Ramírez et al., 2016*).

## Differential expression analysis of repetitive element subfamilies

RNA-seq reads were downloaded from ENA:PRJEB11740, and adapters were trimmed using Trimgalore version 0.5.0 (RRID:SCR_011847). Reads were then aligned and processed using the RepEnrich2 pipeline RRID:SCR_021733 as previously described (*Criscione et al., 2014*). This package sums the number of reads mapping to each repetitive element subfamily. Differentially expressed subfamilies between *Trim28$^{+/D9}$* and wild-type mice were identified using DEseq2 (RRID:SCR_015687) (*Love et al., 2014*) with a p-val threshold of 0.05.

## Acknowledgements

We thank Vanessa Wegert for valuable technical support. We would also like the thank the and members of the Schones lab for helpful comments and suggestions. This work was supported by the National Institutes of Health, grants R01DK112041, R01CA220693 (D.E.S.). The research reported in this publication included work performed in the Integrative Genomics and Analytical Cytometry Cores supported by the National Cancer Institute of the National Institutes of Health under award number P30CA033572.

# Additional information

## Funding

| Funder | Grant reference number | Author |
|---|---|---|
| National Institutes of Health | R01DK112041 | Dustin E Schones |
| National Institutes of Health | R01CA220693 | Dustin E Schones |

The funders had no role in study design, data collection and interpretation, or the decision to submit the work for publication.

## Author contributions

Kevin R Costello, Conceptualization, Data curation, Formal analysis, Investigation, Methodology, Validation, Visualization, Writing - original draft, Writing – review and editing; Amy Leung, Investigation, Methodology, Writing – review and editing; Candi Trac, Michael Lee, Investigation, Writing – review and editing; Mudaser Basam, Investigation, Methodology, Software, Writing – review and editing; J Andrew Pospisilik, Methodology, Resources, Writing – review and editing; Dustin E Schones, Conceptualization, Formal analysis, Funding acquisition, Investigation, Methodology, Project administration, Supervision, Writing - original draft, Writing – review and editing

## Author ORCIDs

Kevin R Costello http://orcid.org/0000-0001-6104-0776
Dustin E Schones http://orcid.org/0000-0001-7692-8583

## Ethics

All animal protocols were in accordance with German and United Kingdom legislation; Project license numbers 80/2098, 80/2497, and 35-9185.81/G-10/94.

## Decision letter and Author response

Decision letter https://doi.org/10.7554/eLife.71104.sa1
Author response https://doi.org/10.7554/eLife.71104.sa2

# Additional files

## Supplementary files

- Transparent reporting form
- Supplementary file 1. Annotation of IAP clades.

## Data availability

All datasets generated in this study have been submitted to GEO under accession code GSE176176.

The following dataset was generated:

| Author(s) | Year | Dataset title | Dataset URL | Database and Identifier |
|---|---|---|---|---|
| Costello KR, Leung A, Trac C, Lee M, Basam M, Pospisilik JA, Schones DE | 2021 | Mechanisms of interindividual epigenetic variability at CpG dense transposable elements | https://www.ncbi.nlm.nih.gov/geo/query/acc.cgi?acc=GSE176176 | NCBI Gene Expression Omnibus, GSE176176 |

The following previously published datasets were used:

| Author(s) | Year | Dataset title | Dataset URL | Database and Identifier |
|---|---|---|---|---|
| Wolf G, de Iaco A, Sun MA, Bruno M, Tinkham M, Hoang D, Mitra AP, Ralls S, Trono D, Macfarlan TS | 2019 | Retrotransposon reactivation and mobilization upon deletions of megabase-scale KRAB zinc finger gene clusters in mice | https://www.ncbi.nlm.nih.gov/geo/query/acc.cgi?acc=GSE115291 | NCBI Gene Expression Omnibus, GSE115291 |
| Gu T, Lin X, Cullen SM, Luo M, Jeong M, Estecio M, Shen J, Hardikar S, Sun D, Su J, Rux D, Guzman A, Lee M, Chen JJ, Kyba M, Huang Y, Chen T, Li W, Goodell MA, Qs Li | 2018 | The role of DNMT3A and TET1 in regulating promoter epigenetic landscapes | https://www.ncbi.nlm.nih.gov/geo/query/acc.cgi?acc=GSE100957 | NCBI Gene Expression Omnibus, GSE100957 |
| Kazachenka A, Bertozzi TM, Sjoberg-Herrera MK, Walker N, Gardner J, Gunning R, Pahita E, Adams S, Adams D, Ferguson-Smith AC | 2017 | The BLUEPRINT Murine Lymphocyte Epigenome Reference Resource [ChIP-seq] | https://www.ncbi.nlm.nih.gov/geo/query/acc.cgi?acc=GSE94658 | NCBI Gene Expression Omnibus, GSM2480410 |
| Thomson JP, Skene PJ, Selfridge J, Guy J, Deaton A, Kerr A, Webb S, Andrews R, James KD, Turner DJ, McLaren S, Illingworth RS, Bird AP | 2010 | Genome-wide maps of CFP1, RNA Polymerase II and H3K4me3 in mouse brain | https://www.ncbi.nlm.nih.gov/geo/query/acc.cgi?acc=GSE18578 | NCBI Gene Expression Omnibus, GSE18578 |
| Li X, Liu Y, Salz T, Hansen KD, Feinberg A | 2016 | Whole genome analysis of the methylome and hydroxymethylome in normal and malignant lung and liver | https://www.ncbi.nlm.nih.gov/geo/query/acc.cgi?acc=GSM1716957 | NCBI Gene Expression Omnibus, GSM1716957 |
| van de Lagemaat LN, Flenley M, Lynch MD, Garrick D, Tomlinson SR, Kranc KR, Vernimmen D | 2018 | ChIP-seq analysis of CFP1 and related molecules | https://www.ncbi.nlm.nih.gov/geo/query/acc.cgi?acc=GSE114084 | NCBI Gene Expression Omnibus, GSM3132538 |
| Li B, Qing T, Zhu J, Wen Z, Yu Y, Fukumura R, Zheng Y, Gondo Y, Shi L | 2017 | A Comprehensive Mouse Transcriptomic BodyMap across 17 Tissues by RNA-seq | https://www.ncbi.nlm.nih.gov/bioproject/?term=PRJNA375882 | NCBI BioProject, PRJNA375882 |
| Dalgaard K, Landgraf K, Heyne S, Lempradl A, Longinotto J, Gossens K, Ruf M, Orthofer M, Strogantsev R, Selvaraj M, Casas E, Teperino R, Surani MA, Zvetkova I, Rimmington D, Tung YC, Lam B, Larder R, Yeo GS, O'Rahilly S, Vavouri T, Whitelaw E, Penninger JM, Jenuwein T, Cheung CL, Ferguson-Smith AC, Coll AP, Körner A, Pospisilik JA, Tt Lu | 2016 | Trim28 Haploinsufficiency Triggers Bi-stable Epigenetic Obesity | https://www.ebi.ac.uk/ena/browser/view/PRJEB11740 | European Nucleotide Archive, PRJEB11740 |

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
