## [Decision Letter]

**Acceptance summary:**

This study aims at understanding how certain retrotransposons escape chromatin-based silencing. Focusing on variably methylated IAP copies (VM-IAPs) in the mouse, the authors show that escapees share sequence variations that alter KRAB zinc finger protein (KZFP) binding and KAP1 recruitment, proximity to expressed genes and high CpG content. Analysis of human retrotransposons in a human KZFP-free mouse cell line recapitulates some of these observations. The authors propose that ZF-CxxC proteins play a role in establishing permissive chromatin at retrotransposons that harbor high CpG content and weak KZFP binding. Although correlative, these findings open the path for further mechanistical demonstration. The paper is of interest to readers in the field of epigenetics, genome biology and transposable elements.

**Decision letter after peer review:**

Thank you for submitting your article "Mechanisms regulating interindividual epigenetic variability at transposable elements" for consideration by *eLife*. Your article has been reviewed by 3 peer reviewers, and the evaluation has been overseen by a Reviewing Editor and Patricia Wittkopp as the Senior Editor. The following individual involved in review of your submission has agreed to reveal their identity: Todd S Macfarlan (Reviewer #4).

Essential revisions:

The reviewers all agreed on the interest of the question and the significance and robustness of the analyses, even though the conclusions may not be entirely novel in some places. They also raised the requirement for 1- providing improved method description, 2-performing additional bioinformatic analyses, 3- not overinterpreting correlative data, 4- discussing previous literature. Please refer to the main three points further expanded below to prepare your revisions, and please consult the detailed points raised by the reviewers, for further modification, editing and discussion in a point-by-point rebuttal.

1) More information is needed regarding the Material and Methods. This is key to understand how the data were treated in terms of bioinformatic analyses: i) precise everywhere it is required whether unique or multiple mapping was used, ii) precise whether 5'LTR and/or 3'LTR were used for LTR alignment when referring to full length VM-IAP copies, iii) considering the strain-specificity of some VM-IAPS and KZFP controllers, when relying on public mouse datasets, specify from which mouse strain they come from. The same mouse strain should ideally be used throughout the study.

2) Additional bioinformatic analyses are needed to strengthen the paper:

– plot KAP1 across IAPLTR2 sequences and over IAPEz internal sequence (reviewer #4).

– in VM versus non-VM comparisons, please use a random set of sequences to compare the same number of sequences (the number of VM-IAPs being overwhelmingly smaller compared to other categories of IAPS (non-VM, others)) (reviewer #3).

– use GTEX human liver methylation instead of the HepG2 cell line WGBS dataset (reviewer #2).

– from their CPF1 CandR in wildtype liver, the authors should analyze the level of CPF1 binding at VM-IAPs specifically, as they did for H3K4me3 (from public B cell dataset).

4 – Data are only correlative as this stage. VM-IAPs are less bound by KAP1/KZFP, but whether this lower recruitment is key to the VM status would need to be genetically tested: by modifying the KZFP binding site of a unique VM-IAP to convert towards weaker or stronger binding. Similarly, whether CFP1 presence on VM-IAPs was not demonstrated here to be causative of the VM-IAP status. IAPs that fail to be protected by KZFPs may just be accessible to all DNA binding proteins. It would require testing whether KAP1 or KZFP loss leads to CFP1 recruitment and then test if CPF1 is required for the VM status in mice. Considering the amount of time and work that these approaches would require, they are not requested within the frame of this revision. However, without formal demonstration, the authors should balance their discussion, and acknowledge that their data provide correlative, not causative, evidence, and should be treated as such. Reviewer #4 listed several instances where the text should be toned down.

*Reviewer #2 (Recommendations for the authors):*

Overall I think this could be a good manuscript if more emphasis was put on precision and an extensive description of patterns observed, as well as a more detailed analysis and a balanced discussion of the CPF1 results and their meaning for our mechanistical understanding of VM-IAPs and other epistable epialles. In addition to the points mentioned in the public review:

– I think the title is not supported by the content of the article and should be modified to something more neutral that showcases the novel findings but does not overplay them. There is no new 'mechanism' in this article, but a few correlations and observations, some of which could suggest future research avenues to prove if they play a mechanistical role. Similarly, claims of the same nature in the abstract, results and Discussion sections should be toned down.

– Much of my problems with the current manuscript have to be with the interpretation of the CPF1 findings – in my view there is absolutely no reason to think there is a mechanistical link between VM-IAP status and CPF1. Derepressed elements being now bound by various factors because they are accessible is not surprising and does not imply any causality with the VM status.

– As described in the public review, a clear description of what strains which dataset comes from is imperative, as KZFPs and young IAPs can be strain specific.

– the 'IAPLTR#' nomenclature should be changed to IAP LTRs #.

– 'euchromatic environment' should be changed to 'proximity of expressed genes' or a similar definition. There can be pockets of local repression close to expressed genes.

– less than 1 kb from an annotated enhancer element… in what cell type(s)? What is the genome distribution of these elements – is there one every 10 kb, which is possible if the database from all cell types was used (which would not be relevant at all)? Some kind of enrichment calculation should be performed instead.

– sentence starting at line 131: I see no support for such a strong statement, even if the above point is clarified. There is enrichment for sure but it is not exclusive, which is implied here with 'must'.

– Figure 1: Why only IAPLTR1s, and not show IAPLTR2s?

– line 137: 'intact' is most probably not correct here – these all have their full open reading frames with no mutations?

– line 142-145: wrong definition of the grey sections – these are gaps, not sequence with no homology.

– provide justification for using only the first 150 bps of IAPEz-ints – repression from KZFPs can have a longer range than a single nucleosome.

– line 181: 'least conserved' is potentially misleading – is it that they were lost in other strains, or that they are new a few related strains? 'conserved' implies the former, while I think here it is the latter. Should be changed elsewhere in the text and in figure legends (for example 2C) to something more precise.

– Figure 2A: IAPEz-clade # should be IAPEz-int clade #.

– Figure 2B: is there no clade 3-4 LTRs associated with IAPLTR2s?

– Figure 2C: again, why no mention of LTR2s?

– Figure 2C: 'cade' instead of 'clade' in a few spots.

– Figure 2C: This one is overall confusing – are clade 2, 3 and 4 never associated with IAPEz-Int? Figure 2B show that they are, and the legend should reflect this, as now this gives the impression that they are solo LTRs given the lack of annotation following the initial presence of it.

– line 224-225: this is backward logic, there's no such demonstration. It should be precised that 'in the absence of repression from KZFPs, IAPs are capable…'

– Figure 3A: 'Mouse ERVs LTRs', in both the figure and the text.

– Figure 3A: The use of the CpG Score metric is puzzling me – why restrict it to 'the 200 bp window with the densest CpG'? This is never explained – why not use the full element?

– line 245: comparing to random sampling of the genome does not make any sense – compare instead to non variably methylated loci of the same TE family (or families).

– Figure 4: be careful in calling 40 million year old elements 'young' – the timescale between mouse and human-specific elements is very different.

– Use GTEx liver methylation instead of in addition to the HepG2 datasets.

– line 285: what tissue was used from the Tc1 mice? Erythroblasts just the same? The expression of KZFPs is also tissue specific, this comparison should be made in ES cells where DNA methylation patterns are established.

– line 322: The sentence is completely unsupported by the data.

– Figure 6B: please compare to the genome-wide distribution of TEs – is it surprising to find that proportion of Mus-specific elements?

– Figure 6D: the use of random sampling of non-responsive TEs is not adequate – compare with random sampling from the same families where you see increased expression.

– The discussion should be rewritten to take into account the above comments – as of now, much of the CPF1 section is not supported by the data.

– provide version numbers for all software and databases used.

– justify the use of bowtie1 in some cases, and bowtie2 in others. Use a single pipeline is preferred.

– Bowtie2 to map RNA-seq is not appropriate as it does not take splicing into account – please use a RNA-seq specific mapping strategy.

*Reviewer #3 (Recommendations for the authors):*

The authors should state the number of copies they are referring to (the VM-IAPs). It seems to me that there is a small number of variably methylated IAPs and this is such a tiny number that reinforces the idea that most IAPs are silenced, and only a few are able to escape silencing. This discussion strengthens the fact that is not only a matter of IAP sequence, but also context, and potentially strain as the authors suggest in the conclusion. Finally, this would put into context the idea that the IAPs are "winning this arms race".

Since the authors are using clusters of IAP sequences, it would be interesting to estimate the age of such clusters and their relationship. This would imply discussing other articles that have previously classified IAP sequences (for instance 10.1002/mc.20576), and even taken into account their methylation level (10.1186/1471-2164-14-48), and potentially their CpG density.

As stated by the authors, 93% of VM-IAPLTR1s are full length sequences, so I wonder how LTRs stemming from the same copy were treated? Are both included in the analysis? Also, of these 93% FL copies, what are the differences in sequence/methylation/binding KZFP/KAP1 etc, between the two LTRs? Are there any FL copy that has different sequences at their LTR and therefore different chromatin state? Previous articles have shown that LTRs bearing the same sequence can have different methylation/chromatin state depending on their distance to nearby genes. Can the authors include this observation in their discussion?

I don't understand the consensus mapping for ChIP-seq for the ZFPs, while the KAP1 ChIP-seq is done on individual IAPLTR1 element (is this uniquely mapping reads?). From the methods, I understand that the ChIP-seq reads are mapped to the mm10 genome, and then the enrichment for each base is translated to a consensus in order to illustrate as a heatmap in Figure 1? Are these uniquely or multimapped reads? These methods should be stated clearly.

In order to associate chromatin environment and VM-IAP state, the authors demonstrate that VM-IAPs are enriched within 50Kb of constitutively expressed genes. This distance seems very large for me, and I wonder how the authors have decided to chose it. Finally, In the Elmer et al., article, where the VM-IAP set is chosen, there is a valid association between CTCF and variably methylated IAP copies. In my opinion this shows how the chromatin environment is associated with VM-IAP state. The authors should discuss this in their manuscript.

The four wild-derived mouse inbred strains from Nellaker et al., ((CAST/EiJ, PWK/PhJ, WSB/EiJ and SPRET/EiJ)) have disproportionate numbers of ERV TEV calls (see Figure 1c from their paper), which could be associated with false positive calls in those species. It would be more stringent to remove these strains from this analysis, in order to have a conservation estimation between lab strains.

Given that VM-IAPs are a smaller set than all the other IAPs, comparisons stating enrichment or "more likely", should take into account these differences. For instance, on Figure 3 D or E, the "other IAPs" or "non-VM" should be a random set that has the same number of copies as the "VM" set.

In the transchromosomic mouse, the authors show that human TE derived CpG islands are hypomethylated due to lack of KZFP that would silence such TEs. I wonder what is the extent of hypomethylation in TEs that are not CpG-rich? It would strengthen the authors argument to show that non-CpG rich TEs are highly methylated (I hope I didn't miss something here though!).

*Reviewer #4 (Recommendations for the authors):*

How the conclusions of this paper could be strengthened:

– To establish that the identified sequence variants are causal to diminished KZFP recruitment and increased CxxC domain containing protein recruitment, motifs could be called in the sequence variants and identified motifs could be deleted and KZFP and CxxX domain containing protein recruitment profiled to assess if the proposed competition between KZFP and CxxC is dependent on the sequence variant.

– As decreased Kap1 recruitment is central to the conclusions of the paper, the authors should provide Kap1 coverage heatmaps for the IAP regions in Figures S1.6 and 2.A. This data could support the conclusion that sequence clades confer altered KAP1 mediated silencing for IAP elements by expanding the observation for IAPLTR1 to IAPLTR2 elements.

– To strengthen the claims of the paper in respect to mouse VM-IAPs, the authors could assess CFP1 binding by ChIP-seq or cutandrun (which they did successfully in human) in previously used available mouse embryonic stem cell lines and mice that lack mouse-specific KZFPs and no longer repress VM-IAPs (see Bertozzi 2020 and Wolf 2020).

– To test if genetic context indeed causes VM, the authors could e.g. remove and ectopically insert a VM-IAP to a different context to test if VM is lost.

– Line 121 'While sequence has been shown to be a factor in the establishment of variable methylation' Please add a reference (Bertozzi et al., for example) to the experimental evidence supporting this claim.

– The authors could consider more directly comparing human and mouse data by performing CFP1 cutandrun or ChIP-seq (that the lab successfully performed in Tc1 liver) also in mouse instead of H3K4me3 as a proxy.

How the presentation of the data could be improved:

– Given that the sequence features distinguishing IAPLTR1 and 2 elements that are or are not variably methylated are a main claim of the paper, the authors should consider providing more reader-friendly sequence information for the observed sequence changes that correlate with diminished KZFP recruitment (e.g. in Figure S1.1 and also for internal seq in S2.1). In the related pre-print, but not the manuscript, one can zoom into the pdf to observe the sequences, but it is very hard to identify actual sequence changes and where they reside. A detailed file in the supplement or zoom ups of the exact sequence changes referred to would be helpful.

– The logic of the sentence in line 154f is not obvious "As the majority of IAPLTR2s exist as solo LTRs this indicates that these elements can be repressed through binding of KZFPs to the IAP internal sequence". Why does it follow from existing as solo LTR that the LTR can be repressed via KZFP binding to the internal sequence? Figure S1.7 does not help understand this point. Please clarify.

– Figure 2C: the authors could consider showing conservation of all VM-sequences individually (given there are not so many) in the supplement in addition to the whole clade.

– In some cases, the manuscript would benefit from toning down statements about causality, when only correlation is assessed (see public review). Further examples include:

– Figure 1 heading "Sequence and chromatin context influence the establishment of a VM-IAPLTRs." The figure shows that VM-IAPs occur in a different chromatin context (less Kap1, more close to constitutively expressed genes and regulatory elements), but not that this context influences variable methylation.

– Relating to the same Figure in 107f: "these results demonstrate that sequence can be a contributing factor in the establishment of VM-IAPs". More appropriate would be "these results suggest that sequence could be a contributing factor in the establishment of VM-IAPs.

– Relating to the same Figure the header in Line 120 'chromatin environment can influence establishment of VM-IAP'. More appropriate would be 'correlates with a distinct chromatin environment'.

– Line 198 'IAPs that have loss of KZFP binding have recruitment of the ZF-CxxC containing proteins TET1 and CFP1' while data supporting this statement is shown in the Tc1 mouse model, the data in this section for mouse is correlative. It does support presence, rather than 'recruitment' of the ZF-CxxC domain containing proteins.

– Line 253f 'our results highlight the potentially conserved importance of high CpG density in the establishment of variable methylation' The human data does not show that CpG density establishes LTR12C elements as variably methylated. It shows that LTR12C elements that are variably methylated are also CpG dense.

---

## [Author Response]

Essential revisions:The reviewers all agreed on the interest of the question and the significance and robustness of the analyses, even though the conclusions may not be entirely novel in some places. They also raised the requirement for 1- providing improved method description, 2-performing additional bioinformatic analyses, 3- not overinterpreting correlative data, 4- discussing previous literature. Please refer to the main three points further expanded below to prepare your revisions, and please consult the detailed points raised by the reviewers, for further modification, editing and discussion in a point-by-point rebuttal.1) More information is needed regarding the Material and Methods. This is key to understand how the data were treated in terms of bioinformatic analyses: i) precise everywhere it is required whether unique or multiple mapping was used, ii) precise whether 5'LTR and/or 3'LTR were used for LTR alignment when referring to full length VM-IAP copies, iii) considering the strain-specificity of some VM-IAPS and KZFP controllers, when relying on public mouse datasets, specify from which mouse strain they come from. The same mouse strain should ideally be used throughout the study.

We thank that the reviewers for raising these points. In the revised manuscript:

1. We have elaborated on unique vs. multimapping approaches for all analyses. Briefly, we used only uniquely mapped reads in all analyses except for the analysis of KAP1 and KZFP binding at different sequence clades (Figure 1). The KAP1 and KZFP ChIP-seq datasets were unable to map uniquely to these highly repetitive loci. However, given that we were interested in determining which KZFPs, and KAP1, can bind to which clades, multimapped reads are appropriate. Bowtie1 with the -m 1 option was used for unique alignments and Bowtie2 with default settings was used where multimapping was required. This was done as bowtie1 has a higher true positive rate for uniquely mapped single end reads than bowtie2 https://www.ncbi.nlm.nih.gov/pmc/articles/PMC6935493/. This has been more thoroughly explained in the methods of the revised manuscript (see page 19, “Alignment of existing ChIP-seq data”).

2. We used both 5’ and 3’ IAPLTRs for all multiple sequence alignments and hierarchical clustering. All elements with the same sequence had the same KZFP binding profile regardless of their position relative to the IAPEz-int. We have attempted to make this more clear in the revised manuscript (see page 5, line 109).

3. We have included more information about the mouse strains used for each dataset. (see page 24, “Availability of data and materials”).

2) Additional bioinformatic analyses are needed to strengthen the paper:– plot KAP1 across IAPLTR2 sequences and over IAPEz internal sequence (reviewer #4).– in VM versus non-VM comparisons, please use a random set of sequences to compare the same number of sequences (the number of VM-IAPs being overwhelmingly smaller compared to other categories of IAPS (non-VM, others)) (reviewer #3).– use GTEX human liver methylation instead of the HepG2 cell line WGBS dataset (reviewer #2).– from their CPF1 CandR in wildtype liver, the authors should analyze the level of CPF1 binding at VM-IAPs specifically, as they did for H3K4me3 (from public B cell dataset).

We thank that the reviewers for suggesting these analyses to strengthen the manuscript. In the revised manuscript:

1. We have reworked Figure 1 to show KAP1/KZFP binding profiles for IAPLTR2s (see Figure 1 —figure supplement 4) and IAPEz-ints (see Figure 1A).

2. We have compared the VM-IAPs to: (1) a random set of loci and (2) a random selection of non VM-IAPLTRs (see Figure 2 —figure supplement 3) and observed that the VM-IAPs have increased H3K4me3 relative to both of these.

3. We have changed the HepG2 methylation data to human liver methylation data (see Figure 4). We observe the same trends with the human liver data as we saw with the HepG2 methylation profiling.

4. We profiled CFP1 binding at VM-IAPs using an existing CFP1 ChIP-seq dataset from C57BL/6 mice (see Figure 2 —figure supplement 2). The CFP1 CUTandRUN profiles that we generated from the WT and *Trim28 D9/+* mouse livers were in an FVB background, so we used the existing data from C57BL/6 mice to keep the strains consistent.

4 – Data are only correlative as this stage. VM-IAPs are less bound by KAP1/KZFP, but whether this lower recruitment is key to the VM status would need to be genetically tested: by modifying the KZFP binding site of a unique VM-IAP to convert towards weaker or stronger binding. Similarly, whether CFP1 presence on VM-IAPs was not demonstrated here to be causative of the VM-IAP status. IAPs that fail to be protected by KZFPs may just be accessible to all DNA binding proteins. It would require testing whether KAP1 or KZFP loss leads to CFP1 recruitment and then test if CPF1 is required for the VM status in mice. Considering the amount of time and work that these approaches would require, they are not requested within the frame of this revision. However, without formal demonstration, the authors should balance their discussion, and acknowledge that their data provide correlative, not causative, evidence, and should be treated as such. Reviewer #4 listed several instances where the text should be toned down.

As suggested by the reviewers, the language has been toned down throughout the paper. We also have added new analyses/figures to examine the role of CpG density and ZF-CxxC proteins more clearly throughout the manuscript and added additional references to frame our results in the context of previous work.

Reviewer #2 (Recommendations for the authors):Overall I think this could be a good manuscript if more emphasis was put on precision and an extensive description of patterns observed, as well as a more detailed analysis and a balanced discussion of the CPF1 results and their meaning for our mechanistical understanding of VM-IAPs and other epistable epialles. In addition to the points mentioned in the public review:– I think the title is not supported by the content of the article and should be modified to something more neutral that showcases the novel findings but does not overplay them. There is no new 'mechanism' in this article, but a few correlations and observations, some of which could suggest future research avenues to prove if they play a mechanistical role. Similarly, claims of the same nature in the abstract, results and Discussion sections should be toned down.

We have adjusted the text and the title accordingly.

– Much of my problems with the current manuscript have to be with the interpretation of the CPF1 findings – in my view there is absolutely no reason to think there is a mechanistical link between VM-IAP status and CPF1. Derepressed elements being now bound by various factors because they are accessible is not surprising and does not imply any causality with the VM status.

We have toned down the language throughout the revised manuscript. Additionally, we have moved some supplemental figures into the main text and added additional analyses that support that fact that VM-IAPs are bound by ZF-CxxC proteins (see Figure 2).

– As described in the public review, a clear description of what strains which dataset comes from is imperative, as KZFPs and young IAPs can be strain specific.

Thank you for this comment. We have included the strain information for all datasets.

– the 'IAPLTR#' nomenclature should be changed to IAP LTRs #.

Thank you for the comment. We chose the IAPLTR# nomenclature to keep our naming consistent with the nomenclature used in Repbase and Dfam, as described in https://www.nature.com/articles/nrg2165-c1 and https://www.dfam.org/family/DF0001789/ summary.

– 'euchromatic environment' should be changed to 'proximity of expressed genes' or a similar definition. There can be pockets of local repression close to expressed genes.

Thank you. We have made this changed our terminology in the revised manuscript.

– less than 1 kb from an annotated enhancer element… in what cell type(s)? What is the genome distribution of these elements – is there one every 10 kb, which is possible if the database from all cell types was used (which would not be relevant at all)? Some kind of enrichment calculation should be performed instead.

We have elaborated on of this work was profiled (see page 20, “Genomic context analysis”). We used annotated enhancers across cell. Additionally, we compared the frequency at which the clade 3 VM-IAPLTR1s and clade 3 non VM-IAPLTR1s were proximal to these expressed genes and enhancers. We observed that the VM-IAPs were more likely to be proximal to these elements compared to other IAPs (Figure 1D). In regards to the window sizes, we have included a figure demonstrating VM-IAPLTR1 clade3 elements proximal to the expressed transcripts and compared this to the non VM-IAPLTR1 clade 3 elements (see Figure 1 —figure supplement 6).

– sentence starting at line 131: I see no support for such a strong statement, even if the above point is clarified. There is enrichment for sure but it is not exclusive, which is implied here with 'must'.

As suggested by the reviewer, the language has been toned down.

– Figure 1: Why only IAPLTR1s, and not show IAPLTR2s?

We have profiled IAPLTR1s and IAPEz-ints in Figure 1, while IAPLTR2s are included in the supplement (see Figure 1 —figure supplements 1,4,5, and 7). This was done because the potential impact of KZFPs on IAPLTR2 elements has been previously explored (Bertozzi et al., 2020).

– line 137: 'intact' is most probably not correct here – these all have their full open reading frames with no mutations?

We thank the reviewer for the suggestion. We have clarified the language to make it clear that we were profiling IAPLTR elements > 300 bps in size.

– line 142-145: wrong definition of the grey sections – these are gaps, not sequence with no homology.

We thank the reviewer for recognizing this mistake. We have changed the terminology to “gap” as suggested.

– provide justification for using only the first 150 bps of IAPEz-ints – repression from KZFPs can have a longer range than a single nucleosome.

We apologize for not making this clear in the initial submission. This window was used because it has been reported that Gm14419 binds to the 5’ most end of IAPEz-ints. We have made this more explicit in the revised manuscript.

– line 181: 'least conserved' is potentially misleading – is it that they were lost in other strains, or that they are new a few related strains? 'conserved' implies the former, while I think here it is the latter. Should be changed elsewhere in the text and in figure legends (for example 2C) to something more precise.

We thank the reviewer for the suggestion. We have clarified that the language and now state that the elements are more/less polymorphic.

– Figure 2A: IAPEz-clade # should be IAPEz-int clade #.

We thank the reviewer for catching this. This has been corrected in the text.

– Figure 2B: is there no clade 3-4 LTRs associated with IAPLTR2s?

We apologize that this was unclear. IAPLTR1 and IAPLTR2 have independent clades. IAPLTR1s were determined to have 4 clades, while IAPLTR2s were determined to have 2 clades. The IAPLTR1 clade 1 elements have distinct sequences compared to IAPLTR2 clade 1 elements (see Figure 1 —figure supplement 1). This figure’s representation has been changed to try to improve clarity (see Figure 1C and Figure 1 —figure supplement 5).

– Figure 2C: again, why no mention of LTR2s?

IAPLTR2s are now included (see Figure 1 —figure supplement 7).

– Figure 2C: 'cade' instead of 'clade' in a few spots.

We thank the reviewer for catching this. This has been corrected in the text.

– Figure 2C: This one is overall confusing – are clade 2, 3 and 4 never associated with IAPEz-Int? Figure 2B show that they are, and the legend should reflect this, as now this gives the impression that they are solo LTRs given the lack of annotation following the initial presence of it.

We apologize that this was unclear in the original version on the manuscript. The updated manuscript has a completely new Figure 1 which more clearly demonstrates the relationship between the LTRs and IAPEz-Int elements (see Figure 1C, 1E).

– line 224-225: this is backward logic, there's no such demonstration. It should be precised that 'in the absence of repression from KZFPs, IAPs are capable…'

We have adjusted both the language of the text and the content of Figure 2. We now include analysis of DNA methylation in cells where TET1 (a ZF-CxxC protein) is present or absent.

– Figure 3A: 'Mouse ERVs LTRs', in both the figure and the text.– Figure 3A: The use of the CpG Score metric is puzzling me – why restrict it to 'the 200 bp window with the densest CpG'? This is never explained – why not use the full element?

We apologize that this was unclear in the original manuscript. The 200 bp window was used to prevent issues caused by TE size. For example, some L1 elements contain a CpG island at their 5’ UTR, but are relatively CpG poor throughout the rest of the element. The 200 bp window size was chosen to match the minimum size of a CpG island (Gardiner-Garden and Frommer, 1987). We have tried to make this clearer in the revised manuscript by adding a section describing CpG profiling in the methods section (see page 21, “Profiling of CpG score”).

– line 245: comparing to random sampling of the genome does not make any sense – compare instead to non variably methylated loci of the same TE family (or families).

We apologize for any confusion. In Figure 3A (to which this line was referring), we were looking to determine whether evolutionarily recent TEs were more likely to be found at variably methylated loci in humans. We determined the observed frequency that variably methylated loci were found at TEs in the genome and compared this to the frequency expected if this distribution was only due to chance. The random sampling of the genome was preformed to determine that expected distribution. We have worked to make this clearer in the text.

– Figure 4: be careful in calling 40 million year old elements 'young' – the timescale between mouse and human-specific elements is very different.

We thank the reviewer for the suggestion. We have adjusted this language from “young” to “evolutionarily recent” throughout the text.

– Use GTEx liver methylation instead of in addition to the HepG2 datasets.

We thank the reviewer for this suggestion. We have changed the methylation data to human liver data. This change did not impact any of our conclusions.

– line 285: what tissue was used from the Tc1 mice? Erythroblasts just the same? The expression of KZFPs is also tissue specific, this comparison should be made in ES cells where DNA methylation patterns are established.

We agree with the reviewer that is potential shortcoming. The CFP1 ChIP-seq data is from human erythroblasts, while the mouse is profiling liver data. However, as the KZFPs are missing from all cells, the lack of methylation should be consistent across tissue types, as seen reported here https://www.ncbi.nlm.nih.gov/pmc/articles/PMC3560060/.­­

– line 322: The sentence is completely unsupported by the data.

We have toned down the language and expanded the data in the figure corresponding to these results. We show that there is hypomethylation only at the CpG dense TEs and that these TEs are the elements that we see specific CFP1 enrichment in the Tc1 mice.

– Figure 6B: please compare to the genome-wide distribution of TEs – is it surprising to find that proportion of Mus-specific elements?

We have included a figure showing the breakdown of TE ages in mm10 genome (see Figure 5 —figure supplement 2).

– Figure 6D: the use of random sampling of non-responsive TEs is not adequate – compare with random sampling from the same families where you see increased expression.

We apologize for any confusion. Each point refers to the total expression of every element in a TE subfamily, and not the expression of an individual TE in the genome. The goal of this figure is to globally profile changes in expression for each TE subfamily, and not to identify novel TE-initiated transcripts. It is therefore not possible to compare responsive TEs to non-responsive TEs of the same subfamily as each data point contains all TEs for the given family. This has been clarified in the methods (see Methods “Differential expression analysis of repetitive element subfamilies”).

– The discussion should be rewritten to take into account the above comments – as of now, much of the CPF1 section is not supported by the data.

We thank the reviewer for the suggestion and have toned down the language throughout the paper.

– provide version numbers for all software and databases used.

We have added version numbers for all software used.

– justify the use of bowtie1 in some cases, and bowtie2 in others. Use a single pipeline is preferred.

We thank the reviewer for the suggestion. Bowtie1 with the -m 1 option was used to allow us to uniquely align the reads to the genome. Bowtie2 with default settings was used in the few instances where multimapping was required. This was done as bowtie1 has a higher true positive rate for uniquely mapped single end reads than bowtie2 https://www.ncbi.nlm.nih.gov/pmc/articles/PMC6935493/. More information has been included in the methods section to clarify which tool was used for processing each dataset (see the updated methods on page 19, “Alignment of existing ChIP-seq data”).

– Bowtie2 to map RNA-seq is not appropriate as it does not take splicing into account – please use a RNA-seq specific mapping strategy.

We thank the reviewer for the comment. This was done as this is part of the RepEnrich pipeline which is used to globally profile all TEs subfamilies. This has been clarified in the method section (see Methods “Differential expression analysis of repetitive element subfamilies”).

Reviewer #3 (Recommendations for the authors):The authors should state the number of copies they are referring to (the VM-IAPs). It seems to me that there is a small number of variably methylated IAPs and this is such a tiny number that reinforces the idea that most IAPs are silenced, and only a few are able to escape silencing. This discussion strengthens the fact that is not only a matter of IAP sequence, but also context, and potentially strain as the authors suggest in the conclusion. Finally, this would put into context the idea that the IAPs are "winning this arms race".

We thank the reviewer for the suggestion. We have altered figures (see Figure 1C and Figure 1—figure supplement 5) to more clearly state the number of IAPs with are variably methylated compared to the total population.

Since the authors are using clusters of IAP sequences, it would be interesting to estimate the age of such clusters and their relationship. This would imply discussing other articles that have previously classified IAP sequences (for instance 10.1002/mc.20576), and even taken into account their methylation level (10.1186/1471-2164-14-48), and potentially their CpG density.

We thank the reviewer for this suggestion and agree that this is an interesting question. Previous studies, such as the suggested paper (10.1002/mc.20576), have observed that “older” TEs have a higher sequence divergence from the consensus sequence. Interestingly, we observed that the clade 3-4 elements have a higher sequence divergence from the consensus IAPLTR1 sequence, but that these elements are more polymorphic and potentially younger.

This is an area of interest for future research. Additionally, we did not observe any major differences in CpG content between the IAPLTR1 sequence clades.

As stated by the authors, 93% of VM-IAPLTR1s are full length sequences, so I wonder how LTRs stemming from the same copy were treated? Are both included in the analysis? Also, of these 93% FL copies, what are the differences in sequence/methylation/binding KZFP/KAP1 etc, between the two LTRs? Are there any FL copy that has different sequences at their LTR and therefore different chromatin state? Previous articles have shown that LTRs bearing the same sequence can have different methylation/chromatin state depending on their distance to nearby genes. Can the authors include this observation in their discussion?

As suggested by the reviewer, we have elaborated on how we treated these elements in the text (see page 5, line 105).

Briefly, we treated 5’ and 3’ LTRs the same during the clustering analysis. We found that both the 5’ and 3’ LTRs have same KZFP binding profile. Additionally, all FL-IAPs have IAPLTRs belonging to the same clades flanking the internal element, so differences in the 5’ and 3’ sequence do not appear to be responsible for any changes in VM status.

I don't understand the consensus mapping for ChIP-seq for the ZFPs, while the KAP1 ChIP-seq is done on individual IAPLTR1 element (is this uniquely mapping reads?). From the methods, I understand that the ChIP-seq reads are mapped to the mm10 genome, and then the enrichment for each base is translated to a consensus in order to illustrate as a heatmap in Figure 1? Are these uniquely or multimapped reads? These methods should be stated clearly.

We apologize for any confusion. We have clarified this in the methods section (see the updated methods on page 19, “Alignment of existing ChIP-seq data”).

While we used uniquely mapped reads when possible, the KAP1 and KZFP ChIP-seq datasets were unable to map uniquely to these highly repetitive TEs. For these datasets, we used multimapped reads to determine if KZFP/KAP1 had the potential to bind to the identified sequence clades.

In order to associate chromatin environment and VM-IAP state, the authors demonstrate that VM-IAPs are enriched within 50Kb of constitutively expressed genes. This distance seems very large for me, and I wonder how the authors have decided to chose it. Finally, In the Elmer et al., article, where the VM-IAP set is chosen, there is a valid association between CTCF and variably methylated IAP copies. In my opinion this shows how the chromatin environment is associated with VM-IAP state. The authors should discuss this in their manuscript.

We thank the reviewer for the suggestion. Regarding the window size, we sampled multiple different window sizes and ultimately selected 50Kb, which covered the greatest number of VM-IAPs. We have included an additional figure to demonstrate that the trends that we observe are consistent regardless of the window size used (see Figure 1 —figure supplement 6).

Additionally, we have included discussion on how Elmer *et al.,* identified an association between CTCF and variable methylation in our main text, as this is an important observation.

The four wild-derived mouse inbred strains from Nellaker et al., (CAST/EiJ, PWK/PhJ, WSB/EiJ and SPRET/EiJ) have disproportionate numbers of ERV TEV calls (see Figure 1c from their paper), which could be associated with false positive calls in those species. It would be more stringent to remove these strains from this analysis, in order to have a conservation estimation between lab strains.

We thank the reviewer for the suggestion. We have removed these outbred strains from our comparison.

Given that VM-IAPs are a smaller set than all the other IAPs, comparisons stating enrichment or "more likely", should take into account these differences. For instance, on Figure 3 D or E, the "other IAPs" or "non-VM" should be a random set that has the same number of copies as the "VM" set.

We thank the reviewer for this suggestion. We have compared our results using both a smaller selection of other IAPs and compared them to a random selection of loci in the genome (Figure 2 —figure supplement 3). We have also moved the H3K4me3 results to the supplement as this has been observed in Bertozzi *et al.,* 2020.

In the transchromosomic mouse, the authors show that human TE derived CpG islands are hypomethylated due to lack of KZFP that would silence such TEs. I wonder what is the extent of hypomethylation in TEs that are not CpG-rich? It would strengthen the authors argument to show that non-CpG rich TEs are highly methylated (I hope I didn't miss something here though!).

We thank the reviewer for this suggestion. We have added the non-CpG rich TEs to the main figure to show that the high CpG density appears to be necessary for the hypomethylation (Figure 4B-C).

Reviewer #4 (Recommendations for the authors):How the conclusions of this paper could be strengthened:– To establish that the identified sequence variants are causal to diminished KZFP recruitment and increased CxxC domain containing protein recruitment, motifs could be called in the sequence variants and identified motifs could be deleted and KZFP and CxxX domain containing protein recruitment profiled to assess if the proposed competition between KZFP and CxxC is dependent on the sequence variant.

We thank the reviewer for this suggestion. To begin to address this point, we have improved our profiling of the IAPLTR sequences. We highlight the region with the greatest KZFP recruitment and compare the sequences for each clade to the previously determined KZFP binding motif (see Figure 1 —figure supplement 3). However, we believe that this is an area for future exploration.

– As decreased Kap1 recruitment is central to the conclusions of the paper, the authors should provide Kap1 coverage heatmaps for the IAP regions in Figures S1.6 and 2.A. This data could support the conclusion that sequence clades confer altered KAP1 mediated silencing for IAP elements by expanding the observation for IAPLTR1 to IAPLTR2 elements.

We thank the reviewer for the suggestion. We have included the KAP1 heatmaps for IAPLTR1, IAPLTR2, and IAPEz-int elements (see Figure 1 and Figure 1 —figure supplement 4).

– To strengthen the claims of the paper in respect to mouse VM-IAPs, the authors could assess CFP1 binding by ChIP-seq or cutandrun (which they did successfully in human) in previously used available mouse embryonic stem cell lines and mice that lack mouse-specific KZFPs and no longer repress VM-IAPs (see Bertozzi 2020 and Wolf 2020).

We thank the reviewer for the suggestion. We agree that it would be interesting to determine whether we can observe an increase of CFP1 recruitment in the absence of KZFP silencing. We believe that this should be further explored in the future.

– To test if genetic context indeed causes VM, the authors could e.g. remove and ectopically insert a VM-IAP to a different context to test if VM is lost.

We agree that this is an interesting idea that could experimentally confirm that location impacts the establishment of a VM-IAP and hope to explore this in the future.

– Line 121 'While sequence has been shown to be a factor in the establishment of variable methylation' Please add a reference (Bertozzi et al., for example) to the experimental evidence supporting this claim.

We apologize for missing this. We have now included this citation.

– The authors could consider more directly comparing human and mouse data by performing CFP1 cutandrun or ChIP-seq (that the lab successfully performed in Tc1 liver) also in mouse instead of H3K4me3 as a proxy.

We thank the reviewer for the suggestion. We profiled existing CFP1 ChIP-seq data from C57BL/6 mice and observed that there is an enrichment of CFP1 signal at the VM-IAPs (see Figure 2 —figure supplement 2). However, we moved the H3K4me3 results to the supplement, as we find the TET1 occupation and the methylation changes in *Tet1* KO mESCs to be more important to the overall story.

How the presentation of the data could be improved:– Given that the sequence features distinguishing IAPLTR1 and 2 elements that are or are not variably methylated are a main claim of the paper, the authors should consider providing more reader-friendly sequence information for the observed sequence changes that correlate with diminished KZFP recruitment (e.g. in Figure S1.1 and also for internal seq in S2.1). In the related pre-print, but not the manuscript, one can zoom into the pdf to observe the sequences, but it is very hard to identify actual sequence changes and where they reside. A detailed file in the supplement or zoom ups of the exact sequence changes referred to would be helpful.

We thank the reviewer for the suggestion. We have included a more reader friendly version of the IAPLTR sequence variants (see Figure 1 —figure supplement 2). We have also included a zoomed in image of the loci with increased KZFP binding (see Figure 1 —figure supplement 3), in addition to the previously annotated KZFP binding motif.

– The logic of the sentence in line 154f is not obvious "As the majority of IAPLTR2s exist as solo LTRs this indicates that these elements can be repressed through binding of KZFPs to the IAP internal sequence". Why does it follow from existing as solo LTR that the LTR can be repressed via KZFP binding to the internal sequence? Figure S1.7 does not help understand this point. Please clarify.

We thank the reviewer for the suggestion. We have reworked this entire section to improve the clarity and flow of the section ( see the section on “IAP sequence influences KZFP recruitment and establishment of variable methylation” in the revised manuscript).

– Figure 2C: the authors could consider showing conservation of all VM-sequences individually (given there are not so many) in the supplement in addition to the whole clade.

We thank the reviewer for the suggestion. This is not included as it has been previously reported in Kazachenka *et al.*, 2018. We were looking to expand upon their work to see if all elements with shared sequence variants were also more likely to be more polymorphic.

– In some cases, the manuscript would benefit from toning down statements about causality, when only correlation is assessed (see public review). Further examples include:– Figure 1 heading "Sequence and chromatin context influence the establishment of a VM-IAPLTRs." The figure shows that VM-IAPs occur in a different chromatin context (less Kap1, more close to constitutively expressed genes and regulatory elements), but not that this context influences variable methylation.– Relating to the same Figure in 107f: "these results demonstrate that sequence can be a contributing factor in the establishment of VM-IAPs". More appropriate would be "these results suggest that sequence could be a contributing factor in the establishment of VM-IAPs.– Relating to the same Figure the header in Line 120 'chromatin environment can influence establishment of VM-IAP'. More appropriate would be 'correlates with a distinct chromatin environment'.– Line 198 'IAPs that have loss of KZFP binding have recruitment of the ZF-CxxC containing proteins TET1 and CFP1' while data supporting this statement is shown in the Tc1 mouse model, the data in this section for mouse is correlative. It does support presence, rather than 'recruitment' of the ZF-CxxC domain containing proteins.– Line 253f 'our results highlight the potentially conserved importance of high CpG density in the establishment of variable methylation' The human data does not show that CpG density establishes LTR12C elements as variably methylated. It shows that LTR12C elements that are variably methylated are also CpG dense.

We thank the reviewer for the suggestions. We have toned down the language throughout the document.